# Supramammillary neurons projecting to the septum regulate dopamine and motivation for environmental interaction in mice

Andrew J. Kesner [1,2], Rick Shin[1], Coleman B. Calva[1], Reuben F. Don[1], Sue Junn[1], Christian T. Potter [1], Leslie A. Ramsey [3], Ahmed F. Abou-Elnaga[1], Christopher G. Cover[4], Dong V. Wang [1], Hanbing Lu[4], Yihong Yang[4] & Satoshi Ikemoto [1✉]

The supramammillary region (SuM) is a posterior hypothalamic structure, known to regulate hippocampal theta oscillations and arousal. However, recent studies reported that the stimulation of SuM neurons with neuroactive chemicals, including substances of abuse, is reinforcing. We conducted experiments to elucidate how SuM neurons mediate such effects. Using optogenetics, we found that the excitation of SuM glutamatergic (GLU) neurons was reinforcing in mice; this effect was relayed by their projections to septal GLU neurons. SuM neurons were active during exploration and approach behavior and diminished activity during sucrose consumption. Consistently, inhibition of SuM neurons disrupted approach responses, but not sucrose consumption. Such functions are similar to those of mesolimbic dopamine neurons. Indeed, the stimulation of SuM-to-septum GLU neurons and septum-to-ventral tegmental area (VTA) GLU neurons activated mesolimbic dopamine neurons. We propose that the supramammillo-septo-VTA pathway regulates arousal that reinforces and energizes behavioral interaction with the environment.

[1] Behavioral Neuroscience Research Branch, Intramural Research Program, National Institute on Drug Abuse, National Institutes of Health, Baltimore, MD, USA. [2] NIH-Johns Hopkins University Graduate Partnership Program, Baltimore, MD, USA. [3] Ex Vivo Electrophysiology Core, Intramural Research Program, National Institute on Drug Abuse, National Institutes of Health, Baltimore, MD, USA. [4] Neuroimaging Research Branch, Intramural Research Program, National Institute on Drug Abuse, National Institutes of Health, Baltimore, MD, USA. ✉email: satoshi.ikemoto@nih.gov

The supramammillary region (SuM) is a posterior hypothalamic structure located just dorsal to the mammillary body (MB) and anterior to the ventral tegmental area (VTA). It robustly projects to much of the septohippocampal formation[1]. Previous studies found that the SuM regulates hippocampal theta oscillations via the medial septum (MS)[2,3], implicating these structures in navigation and memory. More recent studies suggest roles for the SuM in motivation, reinforcement, and substance abuse. Rats learn to self-administer neuroactive chemicals such as AMPA, picrotoxin, and nicotine, a key substance that reinforces tobacco consumption, directly into the SuM[4–6]. Like the reinforcing effects mediated by dopamine (DA)[7], the reinforcing effects of SuM neuron stimulation are accompanied with arousal as indicated by heightened locomotor activity[8]. Moreover, SuM glutamatergic (GLU) neurons are found to drive wakefulness[9]. Interestingly, the stimulation of SuM neurons increases DA concentrations in the ventral striatum (VStr)[4]. These initial findings raise the possibility that in addition to positive reinforcement, SuM neurons may participate in increasing approach behavior, a motivation function that is firmly implicated in DA neuron activity[10,11].

DA signals are important in both reinforcement learning and motivation[12]. Firing rates of DA neurons are known to convey reward-prediction error and the difference between expected and unexpected reward[13], and inactivation of the DA system disrupts reinforcement learning[14]. In addition, increases in VStr DA activity invigorate approach behavior triggered not only by canonical rewards such as food and conditioned stimuli[10], but also non-canonical rewards such as unconditioned flashes of light[15]. Consistently, inactivation of this DA system disrupts approach behavior, including instrumental (conditioned) and exploratory (unconditioned) behavior, and such effects can occur without disrupting food intake[16–20], suggesting that DA is more important during approach than consummatory behavior. Consistent with DA's role in both reinforcement and motivation, contingent stimulation of DA neurons reinforces responses as shown by intracranial self-stimulation procedures[21,22]. Such a role was initially attributed to a subset of VTA-to-VStr DA neurons[23,24], consistent with the notion of functional heterogeneity among DA neurons[25,26]. However, more recent studies show that contingent stimulation of DA neurons of the substantia nigra reinforces behavior similar to that of VTA DA neurons[21,27], suggesting some common functions shared among many DA neurons.

We conducted a series of operant self-stimulation experiments, which suggest that optogenetic stimulation of SuM GLU neurons projecting to MS GLU neurons and MS GLU neurons projecting to the VTA reinforces behavior. We then used in vivo unit recording and chemical inhibition to examine functional roles of SuM neurons in approach and consummatory responses reinforced by sucrose reward. In addition, we conducted fMRI, fiber photometry, and pharmacological experiments to establish the interactions between the SuM-to-septum-to-VTA GLU pathways and the VTA-to-VStr DA pathway.

## Results

**Phasic excitation of SuM neurons is reinforcing.** We examined whether phasic excitation of SuM neurons is reinforcing using an optogenetic intracranial self-stimulation (ICSS) procedure with channelrhodopsin-2 (ChR2) expressed in SuM neurons of wild-type (WT; C57BL/6 J) mice (Fig. 1a, b and Supplementary Fig. 1). To address site-specificity, we prepared additional groups with viral vectors injected into the SuM with optic fibers placed adjacent to SuM in the MB or unilaterally the VTA. ICSS was measured in an operant-conditioning chamber equipped with two levers (Fig. 1c). One lever was assigned as the 'active' lever on

which a single depression (i.e., fixed ratio of 1) delivered a train of photostimulation (15 pulses at 25 Hz), and the other was the 'inactive' lever, which had no programed consequence. For the first 12 sessions (30 min per session), we examined whether experimentally naive mice would acquire, extinguish, and reacquire lever pressing reinforced by photostimulation. The first 2 sessions were a baseline period where lever pressing was not reinforced by photostimulation, and the three groups did not differ on active-lever presses (Fig. 1d; $3_{group} \times 2_{session}$ ANOVA: $F_{group}(2,19) = 0.55$, $p = 0.5845$). When active-lever pressing was reinforced by photostimulation in sessions 3–7, the mice increased active-lever pressing, but not inactive-lever pressing. The mice with optic fibers targeting the SuM had significantly greater active-lever presses than the mice with optic fibers targeting the MB or VTA, and MB and VTA mice did not differ from one another ($3_{group} \times 5_{session}$ ANOVA: $F_{group}(2,19) = 15.79$, $p < 0.0001$). The SuM-stimulation mice decreased active-lever pressing in sessions 8–10 when lever pressing was not reinforced, and reinstated it in sessions 11 and 12 when the contingent reinforcement returned (Fig. 1d; $3_{group} \times 2_{reinforcer} \times 2_{session}$ ANOVA for sessions 9–12, significant group × reinforcer interaction; $F_{interaction}(2,19) = 9.69$, $p < 0.01$).

To provide additional evidence for reinforcement, we reversed the active–inactive lever assignments for all mice. The SuM-stimulation mice quickly learned to reverse pressing between the two levers when the formerly active lever became inactive and the formerly inactive lever became active (Fig. 1e; $2_{lever} \times 2_{assignment} \times 2_{session}$ ANOVA: significant lever × assignment interaction; $F_{interaction}(1,9) = 10.09$, $p < 0.05$). Conversely, the MB-stimulation and VTA-stimulation mice failed to reverse pressing between the two levers (MB: $F_{interaction}(1,5) = 2.1$, $p = 0.2$; VTA: $F_{interaction}(1,9) = 4.62$, $p = 0.08$). These observations further support that direct stimulation of SuM neurons is reinforcing.

**Excitation of SuM GLU neurons is reinforcing.** The SuM is comprised of a heterogeneous population of neurons including GLU, DA, and GABAergic neurons[3,9,28,29]. To determine which type of neurons are responsible for the reinforcing effect, we selectively expressed ChR2 in these three respective neuron subtypes (Fig. 1f, g and Supplementary Fig. 2). To address site-specificity, we prepared a second cohort of vGlut2-Cre mice injected with a small volume (50 nL, instead of 200 nL) of the vector ($n = 6$). Regardless of injection volumes, both vGlut2-Cre groups quickly learned to press on the active lever with no group difference (Fig. 1h; a $2_{group} \times 2_{lever} \times 7_{session}$ ANOVA, $F_{group}(1,16) = 0.29$, $p = 0.59$); thus, these data were combined. By contrast, TH-Cre and vGat-Cre mice failed to significantly acquire active-lever pressing (a $3_{group} \times 5_{session}$ ANOVA; $F_{group}(2,33) = 5.79$). These results suggest that the excitation of SuM GLU, but not DA or GABA, neurons is reinforcing.

**SuM GLU neurons projecting to the septum and to the paraventricular thalamic nucleus mediate positive and negative reinforcements, respectively.** To determine which downstream region receives reward signals from SuM, we examined three brain regions that receive prominent innervation from SuM GLU neurons and are known to be involved in motivation and affect the septum, the ventral subiculum (vSub), and the paraventricular thalamic nucleus (PVT). Each vGlut2-Cre mouse received a Cre-dependent ChR2 vector into the SuM, and optic fibers into the septum and vSub or the septum and PVT through which they would receive photostimulation to one region at a time. Mice quickly learned to increase lever pressing when they received contingent stimulation at the septum. By contrast, they did not increase lever pressing when they received stimulation at the vSub or PVT

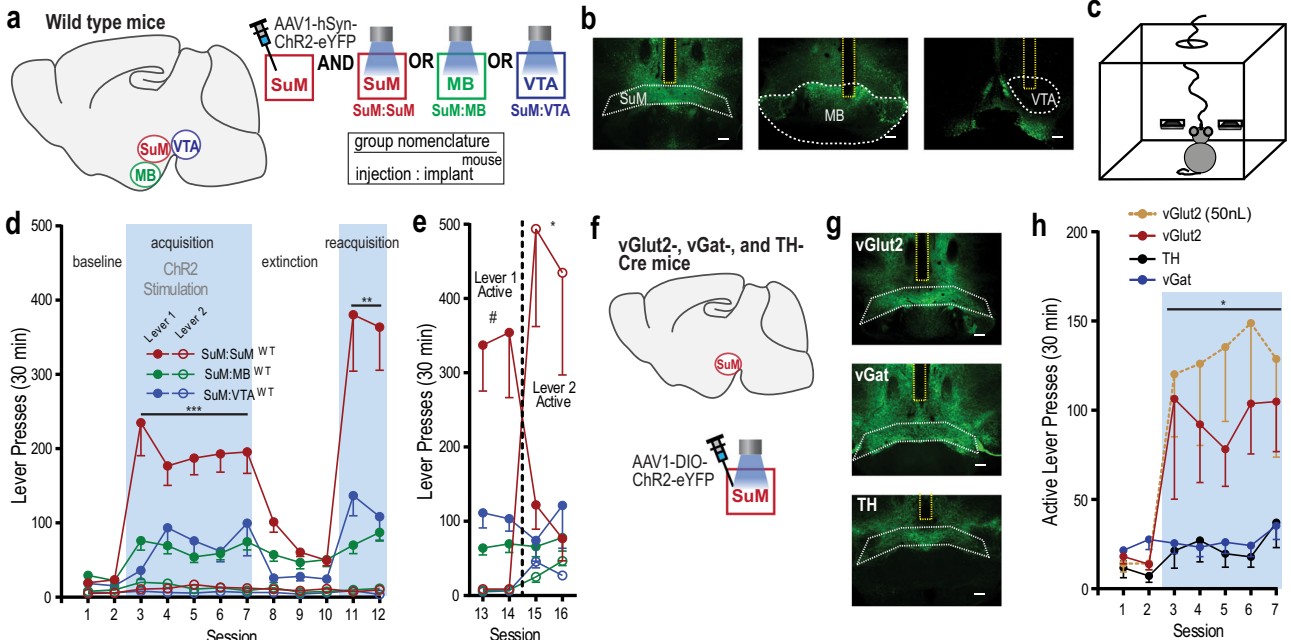

**Fig. 1 Phasic excitation of SuM GLU neurons is reinforcing. a** Schematic of hSyn-ChR2 viral injection targeting SuM neuron cell bodies and optic fiber implantation targeting the SuM, MB, or VTA placements. The group nomenclature box describes how injection site and stimulation site are abbreviated (e.g., SuM:SuM). **b** Examples of histology showing ChR2-eYFP expression in SuM and surrounding tissues with optic fiber placements (yellow dotted) in SuM (top), MB (middle), and VTA (bottom). Scale bar = 200 μm. See Supplementary Fig. 1 for histological results of all mice (n = 22). **c** Schematic of mouse operant-conditioning chamber equipped with two levers. **d** Active (closed circles) and inactive (open circles) lever presses per session (mean − SEM). SuM-stimulation group has significantly higher ICSS rates than the other two groups during acquisition sessions 3–7 and reacquisition sessions 11–12. ***$p < 0.001$ and **$p < 0.01$, with Tukey post hoc HSD test, SuM:SuM$^{WT}$ (n = 10), SuM:MB$^{WT}$ (n = 6), and SuM:VTA$^{WT}$ (n = 6). All animals pressed very little on inactive lever, so their data are shown but not further analyzed. **e** Lever reversal experiment. The data are mean lever presses − SEMs. SuM-stimulation group responded on lever 1 significantly more than on lever 2 when lever 1 was active, #$p < 0.012$; and they reversed responding and responded on lever 2 significantly more than on lever 1 when lever 2 became active, *$p < 0.001$ (Tukey HSD post hoc test). See (**d**) for the n of each group. **f** Schematic showing Cre-specific ChR2 viral injection targeting SuM neurons and optic fiber implantation targeting SuM performed in vGlut2-Cre, TH-Cre, and vGat-Cre mice. **g** Examples of histology showing ChR2-eYFP expression in SuM in vGlut2-Cre (top), TH-Cre (middle), and vGat-Cre mice with optic fiber tracts (yellow) in SuM. Scale bar = 200 μm. See Supplementary Fig. 2 for histological results of all mice (n = 46). **h** Active lever-presses per session (mean − SEM). vGlut2-Cre group (n = 18) responded greater than TH- or vGat-Cre group (n = 7 and 11, respectively), $p_{vGlut2 \ vs \ TH} = 0.037$, $p_{vGlut2 \ vs \ vGat} = 0.016$, $p_{vGat \ vs \ TH} = 0.99$ (Tukey HSD post hoc test). vGlut2-Cre groups consisted of the mice (n = 6) that received a 50 nL volume of the AAV to minimize the diffusion of the vector and the mice (n = 12) that received 200 nL. Source data are provided as a Source Data file.

(Fig. 2a). Because septum placements in our first cohorts were confined to the dorsal part of the septum, and the SuM sends projections to the entire septal complex[1], we prepared a third cohort of mice that had a single implant targeting the ventral MS and the diagonal band of Broca (DBB). Overall mice with optic fibers in the corpus callosum or ventral to the septum near the anterior commissure did not lever press and were not included in any analysis (Supplementary Fig. 3). In addition, the stimulation of VTA GLU neurons projecting to the septum (n = 6) failed to reinforce lever pressing (Fig. 2b; a $4_{group} \times 5_{session}$ ANOVA: $F_{group}(3,45) = 12.70$, $p < 0.0001$). To confirm that terminal stimulation in vSub or PVT had activated local neurons, photostimulation was delivered to these regions 90 min before euthanasia, and brain tissue was stained for c-Fos, a marker for strong neuronal activation (Fig. 2c). c-Fos counts in the septum and PVT were significantly greater when stimulated than when not (Fig. 2d). In the vSub, c-Fos counts were significantly greater in the hemisphere ipsilateral to stimulation (a $2_{stimulation} \times 2_{hemisphere}$ ANOVA: $F_{interaction}(1,13) = 6.578$, $p < 0.05$). Therefore, these results suggest that the stimulation of SuM GLU neurons projecting to the septum is more reinforcing than the other SuM pathways.

It was peculiar to find that stimulation of the SuM-to-Sept GLU pathway supported higher ICSS rates compared to SuM GLU cell-body stimulation (cf. Fig. 1h vs 2b; and Supplementary Fig. 4),

since cell-body stimulation should activate neural populations more efficiently than that of terminal fibers. Furthermore, a growing body of literature suggests that PVT neurons mediate aversion[30–32]. Therefore, the stimulation of SuM GLU cell bodies may have produced mixed effects of reinforcement: a positive effect via the septum and a possible negative effect via the PVT (Fig. 2e). To test this possibility, we used a real-time place preference/aversion procedure (Fig. 2f). Mice reduced the time spent in the stimulation compartment when photostimulation was delivered at the PVT, whereas the same mice increased the time spent in the stimulation compartment when photostimulation was delivered at the septum (Fig. 2g, h; $2_{region} \times 2_{stimulation}$ RM-ANOVA; $F_{interaction}(1,3) = 548.7$, $p < 0.001$). Thus, the results indicate that SuM GLU neurons projecting to the septum mediate positive reinforcement, whereas those projecting to the PVT mediate negative reinforcement.

**Administration of AMPA into the septum is reinforcing and increases approach behavior.** Since stimulation of SuM GLU terminals in the septum is reinforcing, we used an intracranial self-administration procedure (ICSA)[33] to determine whether the direct stimulation of septal neurons via AMPA receptors is reinforcing and, if so, which septal sub-regions mediate such

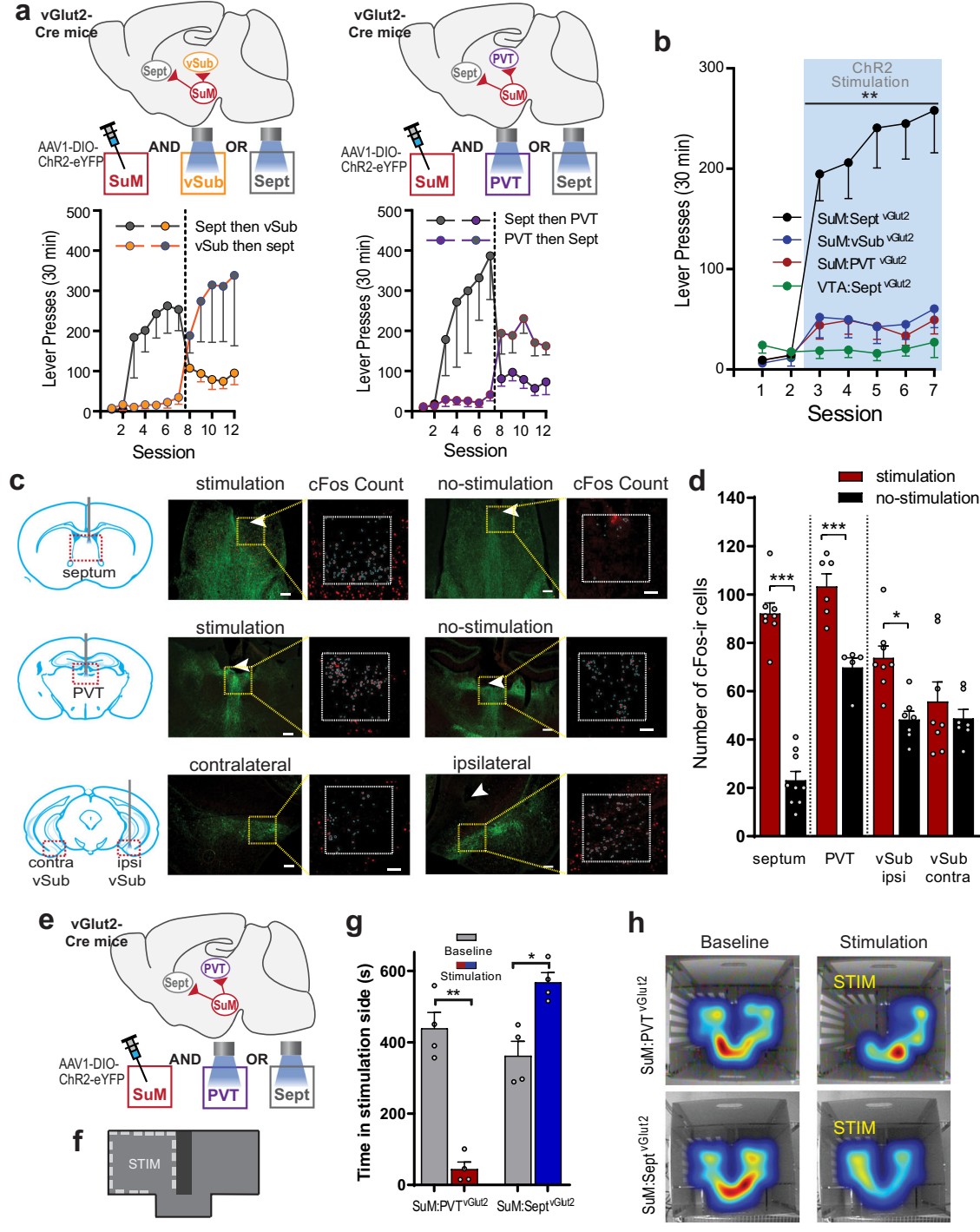

effect. ICSA was performed in rats because rats offer larger brain areas for surveying site-specificity than mice and because the head mounted microinjection pump[34] is too large in mice. Each rat ($n = 111$) received surgery for the implantation of a guide cannula targeting a septal site. While some rats had the opportunity to self-administer AMPA solutions (0.01–0.25 mM in 100 nL per infusion) into a single site, others had the opportunity to self-administer AMPA (0.03–0.1 mM in 100 nL per infusion) into multiple sites by systematically increasing the length of injection cannula over multiple sessions (Fig. 3a, b). Rats quickly learned to self-administer AMPA reliably into the zone along the midline of the septal area, but not the lateral part of the lateral septum (Fig. 3a). AMPA ICSA was completely abolished by co-

administration of the AMPA receptor antagonist ZK 200775 (Fig. 3c; one-way RM ANOVA, $F_{(2,22)} = 16.04$, $p < 0.005$). Moreover, rats did not learn to self-administer AMPA into the nucleus accumbens shell or DBB. These results suggest that the medial part of the septum is particularly important in mediating the reinforcing effect induced by AMPA receptor stimulation. Interestingly, systemic injection of the DA receptor (DAR) antagonist SCH23390 significantly reduced intra-septal AMPA ICSA (Fig. 3d; RM-ANOVA; $F_{(1,6)} = 21.701$, $p < 0.01$), suggesting that the self-administration behavior triggered by intra-septal AMPA depends on DA transmission.

Brief presentation of an unconditioned, visual stimulus (VS) is known to serve as a reward in rodents[35–37]. We took advantage of

**Fig. 2 SuM GLU neurons projecting to the septum and to the paraventricular thalamic nucleus mediate positive and negative reinforcements, respectively. a** Schematics showing Cre-specific virus injections into SuM and optogenetic stimulation sites in Sept ($n = 23$ mice) vs vSub ($n = 7$ mice) or Sept vs PVT ($n = 12$ mice). Data are mean lever presses with SEM for each group. Gray, orange, and purple circles indicate data with septal, vSub, and PVT stimulation, respectively. **b** Summary of active lever-presses (mean − SEM) per session. See (**a**) for procedure and $n$'s. In addition, AAV was injected into VTA with fiber placed in septum ($n = 6$ mice). **p's < 0.005, SuM:Sept^vGlut2 responding more than any other groups over sessions 3–7, Tukey HSD post hoc test. **c** c-Fos expression after stimulation at respective sites. Arrows indicate the tips of optic fiber; green labels indicate vGlut2 neurons expressing ChR2 (scale bar = 200 μm); and c-Fos-ir is shown in red (scale bar = 100 μm). **d** c-Fos counts (mean + SEM). c-Fos counts in septum (stimulation, $n = 8$ mice; non-stimulation, $n = 9$ mice) and PVT (stimulation, $n = 6$; non-stimulation, $n = 5$) were significantly greater when stimulated than when not (unpaired $t$-test, ***p's < 0.001). In the vSub, c-Fos counts were significantly greater in the hemisphere ipsilateral to stimulation than contralateral to stimulation ($n = 8$ mice) or when not stimulated ($n = 7$ mice). *p < 0.001, Bonferroni test. **e** Schematic of optogenetic preparation of within-subject comparison of SuM:Sept^vGlut2 vs SuM:PVT^vGlut2 pathways in real-time place preference. **f** Schematic of real-time place preference chamber. The stimulation compartment (STM) is shown with a dotted line. **g** Time spent in stimulation side ($n = 8$ mice; mean + SEM) of the real-time place preference/aversion chamber during baseline (no stimulation) session and stimulation session with septal or PVT stimulation. **p < 0.001, *p < 0.01, Bonferroni test. **h** Example heatmaps of mouse position showing time spent in each compartment. Warmer colors indicate more time spent in that area. Source data are provided as a Source Data file.

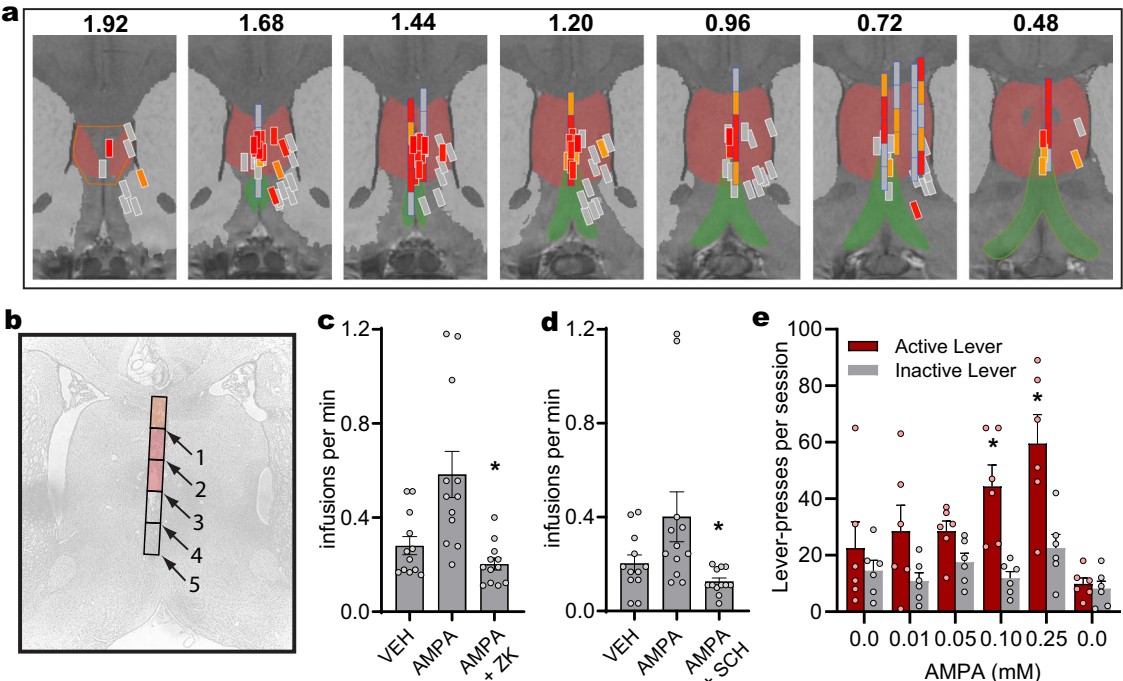

**Fig. 3 Administration of AMPA into the septum is reinforcing and increases approach behavior. a** Effectiveness of AMPA self-administration sites. The septal area is indicated in maroon; the diagonal band of Broca is indicated in green; the striatum is indicated in light gray. Injection sites are shown in rectangles with three different colors, which indicate the effectiveness of self-administration at each injection site (observations from 111 rats): high rate (red; when the highest injection rate is 0.67 injections/min or greater); medium rate (orange; when the site supported 0.46 injections/min in 2 sessions or more); and low rates (gray; when rates fall into all other cases). The numbers indicate distances from bregma. The rat coronal sections are adapted and modified from Waxholm Space rat brain atlas v3 (https://ebrains.eu/service/rat-brain-atlas). **b** Photomicrogram showing a cannula track with estimated injection sites. This rat received AMPA injections at 5 sites along the dorso-ventral track (black squares), and self-administered AMPA at a medium rate with cannula 1, high rates with cannulae 2 and 3, and low rates with cannulae 4 and 5. **c** Effect of the AMPA receptor antagonist on AMPA self-administration. Rats received vehicle, 0.25 mM AMPA alone, and a mixture of 0.1 mM ZK200775 and 0.25 mM AMPA over 3 consecutive sessions. Co-administration of ZK with AMPA significantly reduced self-administration ($n = 12$ rats; mean ± SEM; *p < 0.001, compared to the AMPA value, Tukey HSD post hoc test). **d** Effect of the DA receptor antagonist SCH 23390. ICSA rates of AMPA into the septum is significantly greater than those of vehicle (aCSF) or AMPA with the pretreatment of SCH 23390 ($n = 12$ rats; 0.025 mg/kg, i.p.; mean ± SEM; *p < 0.01, compared to the AMPA value, Tukey HSD post hoc test). **e** Effects of non-contingent intra-septal AMPA injections on visual-stimulus seeking behavior ($n = 6$ rats; mean + SEM; *p's < 0.01, compared to the inactive-lever value, Tukey HSD post hoc test).

the properties of VS to investigate whether increased AMPA transmission in the septum can facilitate on-going seeking behavior reinforced by VS, i.e., visual-sensation seeking. We used a VS because canonically used food rewards elicit consummatory responses, which suppress approach-behavior and can confound interpretation of modulatory effects of a manipulation. When 0.1 or 0.25 mM AMPA was infused into the midline septum in a non-

contingent manner, rats increased active lever pressing reinforced by the 1 s presentation of a VS, while inactive lever pressing, which did not produce a VS, remained unchanged (Fig. 3e; $6_{concentration} \times 2_{lever}$ repeated measures ANOVA; $F_{interaction}(5,25) = 8.71$, $p < 0.0001$). These results suggest that the stimulation of septal neurons via AMPA receptors is not only reinforcing, but also potentiates ongoing approach behavior.

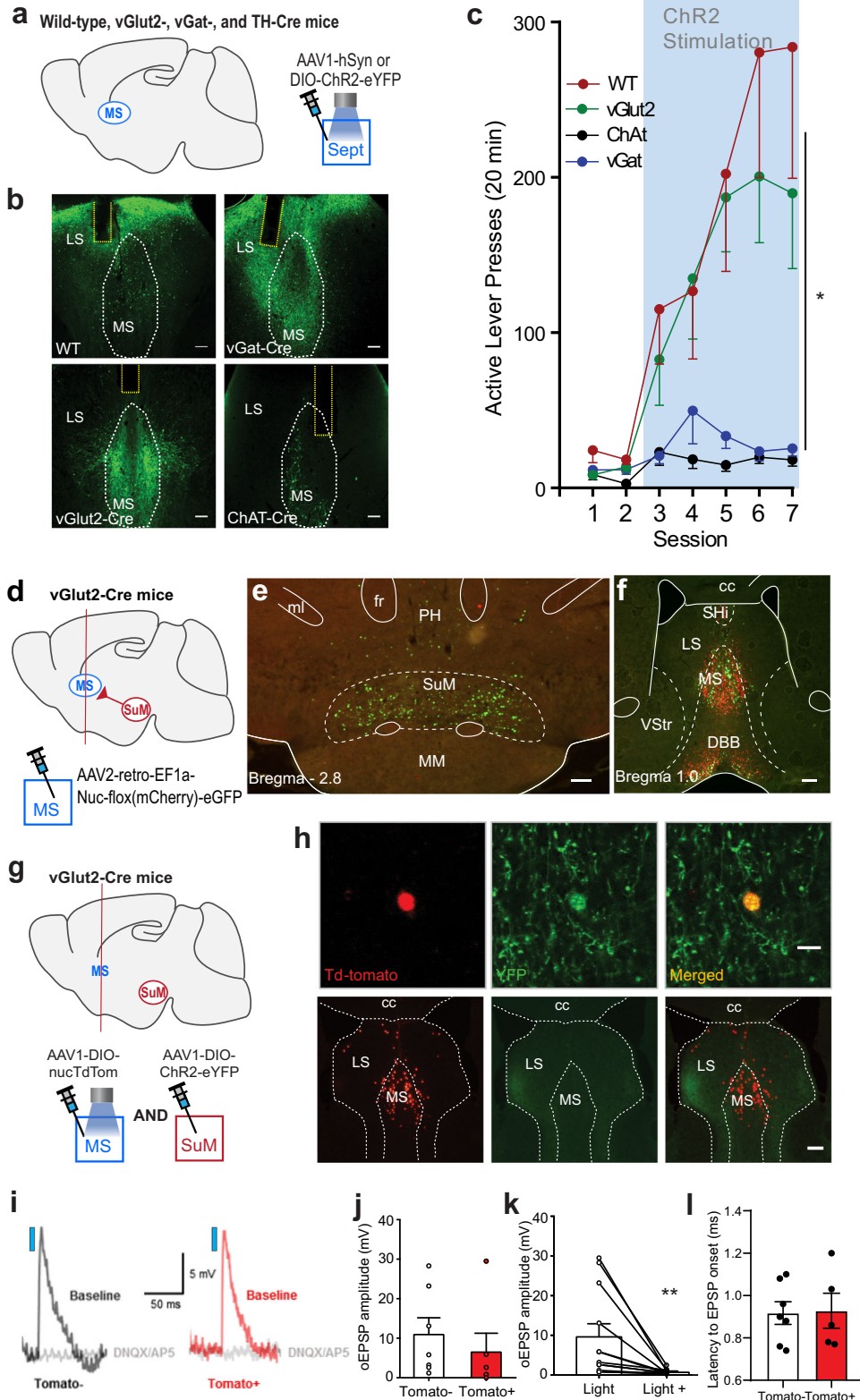

**Excitation of septal GLU neurons is reinforcing**. The above experiment with AMPA injections confirmed that GLU inputs to the septum mediate motivation and reinforcement. Subsequently, we used optogenetic procedures in mice to understand which neurons are responsible for glutamate-mediated positive reinforcement. Here, we first found that nonselective excitation of septal neurons is reinforcing in WT (Fig. 4a–c). Then, we

determined what neuronal phenotype is responsible for the effect. The septum contains GLU, GABAergic, and cholinergic neurons: GLU and cholinergic neurons are primarily located in the MS, while GABAergic neurons are found throughout the septum[38,39]. We found that vGlut2-Cre, but not vGat-Cre or ChAt-Cre, mice increased active lever-presses (Fig. 4c and Supplementary Fig. 5; $4_{group} \times 5_{session}$ ANOVA: $F_{interaction}(12,168) = 4.03$, $p < 0.001$).

**Fig. 4 Excitation of septal GLU neurons is reinforcing and SuM neurons monosynaptically excites septal GLU neurons. a** Schematic showing septal site of AAV injection and optic fiber placement in WT, vGat-, vGlut2-, and ChAt-Cre mice. **b** Example histology of ChR2-eYFP expression in the septum and optic fiber placements (yellow dotted outline) for the 4 strains of mice used. Scale bar = 100 μm. See Supplementary Fig. 5 for all results (n = 48). **c** Active lever-press (mean − SEM) per session per group: WT (n = 10), vGat- (n = 18), vGlut2- (n = 12), and ChAt-Cre (n = 6) mice. $^{*}p_{\text{WT vs vGat}} = 0.0008$, $p_{\text{WT vs ChAt}} = 0.0071$, and $p_{\text{vGlut2 vs vGat}} = 0.0077$ and $p_{\text{vGlut2 vs ChAt}} = 0.0397$ (Tukey post hoc HSD test). **d** Schematic showing an injection of the retrograde vector into the MS of the vGlut2-Cre mouse. **e, f** Photomicrogram showing retrogradely labeled glutamatergic (GLU) cells in green and non-GLU cells in red. n = 4 mice. Scale bars = 200 μm. Abbreviations: cc, corpus callosum; DBB, diagonal band of Broca; fr, fasciculus retroflexus; LS, lateral septum; ml, medial lemniscus; MM, medial mammillary nucleus; MS, medial septum; PH, posterior hypothalamic region; SHi, septohippocampal nucleus; VStr, ventral striatum. **g** Schematic showing injection preparation for ex vivo brain slice electrophysiology experiments to determine whether SuM vGlut2 neurons can monosynaptically excite septal vGlut2 neurons. **h** Top row shows an example of a septum neuron expressing Td-tomato, and therefore glutamatergic, surrounded by ChR2-eYFP fibers, the image was taken before patch-clamp electrophysiology of this neuron. Bottom row shows example histology of the septal area with Td-tomato positive cells localized primarily to MS and SuM:Sept$^{\text{vGlut2}}$ terminals innervating the septal complex. Scale bar top = 10 μm, bottom = 200 μm. **i** Average EPSP traces from recorded septum neurons. Traces are calculated from average responses of cells at the midpoint of data sets. Postsynaptic response is abolished by GLU antagonists. Blue rectangle represents approximately 2 ms blue-light laser stimulation. **j** EPSP amplitude data of septum neurons. Data are mean + SEM with individual data points (circles; 2–4 cells per mouse, n = 4). At least 1 Tomato + and − was recorded per mouse. **k** Effect of GLU antagonists on EPSP amplitude. Data are mean − SEM with individual data points. $^{**}p = 0.01$, paired t-test. See (**j**) for n's. **l** EPSP latency responses after the onset of blue-light pulse. Data are mean ± SEM with individual data points. See (**j**) for n's. Source data are provided as a Source Data file.

The results suggest that the excitation of septal GLU neurons is reinforcing.

**The SuM-to-MS pathway is glutamatergic.** We examined the extent to which SuM-to-MS neurons are glutamatergic. We used a retrograde tracer, AAV2-retro-EF1a-Nuc-flox(mCherry)-eGFP, which results in retrograde eGFP labeling in Cre-containing neurons and mCherry labeling in neurons without Cre (n = 4). Injections of the vector into the MS of vGlut2-Cre mice (Fig. 4d) resulted in retrogradely eGFP-labeled, but not mCherry-labeled, cells densely distributed in the SuM, sparsely distributed in the posterior hypothalamic region, and absent in the mammillary region (Fig. 4e). The same injection produced both eGFP and mCherry labeling at the injection sites as well as in the medial horizontal limb of the DBB (Fig. 4f), which is known to project to the MS. The results suggest that the SuM-to-MS pathway is primarily glutamatergic.

**SuM neurons monosynaptically excite septal GLU neurons.** We performed a slice patch-clamp electrophysiology procedure to determine whether the afferents from the SuM provide excitatory inputs to MS GLU neurons via GLU transmission. We expressed ChR2 in SuM GLU neurons and labeled MS GLU neurons with AAV1-DIO-nucTdTomato for subsequent stimulation and recording (Fig. 4g). All recorded neurons were adjacent to SuM terminals expressing ChR2-eYFP (Fig. 4h). Of the 14 cells recorded, 12, including both tdTomato-positive and tdTomato-negative neurons, displayed an optogenetically induced excitatory postsynaptic potential (oEPSP) upon 2-ms photostimulation (Fig. 4i). There was no significant difference in the magnitude of the EPSP between the two groups (unpaired t-test p = 0.49) (Fig. 4j). In addition, a cocktail of the GLU antagonists DNQX and AP5 abolished the oEPSPs (Fig. 4k). Finally, mean latency to evoke oEPSP from the onset of photostimulation was 0.92 ms (SD = ±0.15; Fig. 4l). It is not easy to compare the latency results with those of other studies because many factors can affect latencies such as recording temperature. Our observed latencies are in line with previous results showing that monosynaptic connections have a response latency of 1.1 ms with electrical stimulation[40], and 2.0 ms with optogenetic stimulation[41]. The short latency to trigger oEPSPs suggests that SuM neurons monosynaptically excite MS GLU neurons via GLU transmission. In addition, these results are consistent with the finding that MS GLU neurons receive synaptic contacts from SuM neurons as measured by monosynaptic pseudorabies tracing[42].

**Consummatory behavior decreases the activity of SuM neurons.** We next performed in vivo extracellular single-unit recordings to examine the activity of SuM neurons during motivated behavior (Fig. 5a). Mice were trained to respond on a lever that led to the presentation of auditory stimuli signaling whether or not sucrose-solution reward would be delivered (Fig. 5b). Over 5 sessions, mice displayed increased latency to enter the sucrose port upon presentation of an auditory stimulus signaling no reward (CS−), while maintaining relatively short latency when presented with the stimulus signaling the reward (CS+) (Fig. 5c; $2_{\text{cue}} \times 5_{\text{session}}$ RM-ANOVA; $F_{\text{interaction}}(4,20) = 4.37$; $p = 0.01$), suggesting that the mice learned to discriminate CS+ from CS−. The mice also spent more time in the sucrose port upon CS+ presentation (mean ± SD times spent 14.8 ± 7.1 s) compared to CS− (3.1 ± 3.3 s), reflecting a consummatory response and no response, respectively. When the mice displayed stable responses between sessions, unit recording sessions started. The most striking observation was that SuM neurons uniformly and markedly decreased firing rates when the mice entered the sucrose port following CS+ compared to CS− and remained low until mice exited from the port (Fig. 5d–f). This suggests that consummatory behavior suppresses SuM neuron activity.

This observation is reminiscent of hippocampal theta oscillations, which occur when the host interacts with the environment and diminish when it engages in self-care activities such as feeding, drinking, or grooming[43,44], and this is not surprising given the previous findings that the SuM modulates hippocampal theta oscillations[2,3]. We examined whether the activity of SuM neurons coincided with locomotion. We found that the activity of 29.6% (45/152) of the neurons was significantly correlated (correlation coefficient R-value > 0 and p-value < 0.01) with locomotor activity during the session. We further analyzed the unit locomotor activity data, with respect to the mice's location within the chamber, based on the rationale that if SuM neurons are involved in reward-seeking process, correlations may change depending on the location within the chamber ('near' vs 'away' from the port; Supplementary Fig. 6a, b). After excluding locomotor and single-unit data during the period that the mice were immobile at the port, we found decreased correlations between the unit activity and locomotor activity and no difference in such correlation between the compartments: 14.4% (22/152) and 16.4% (25/152) for 'near' and 'away', respectively

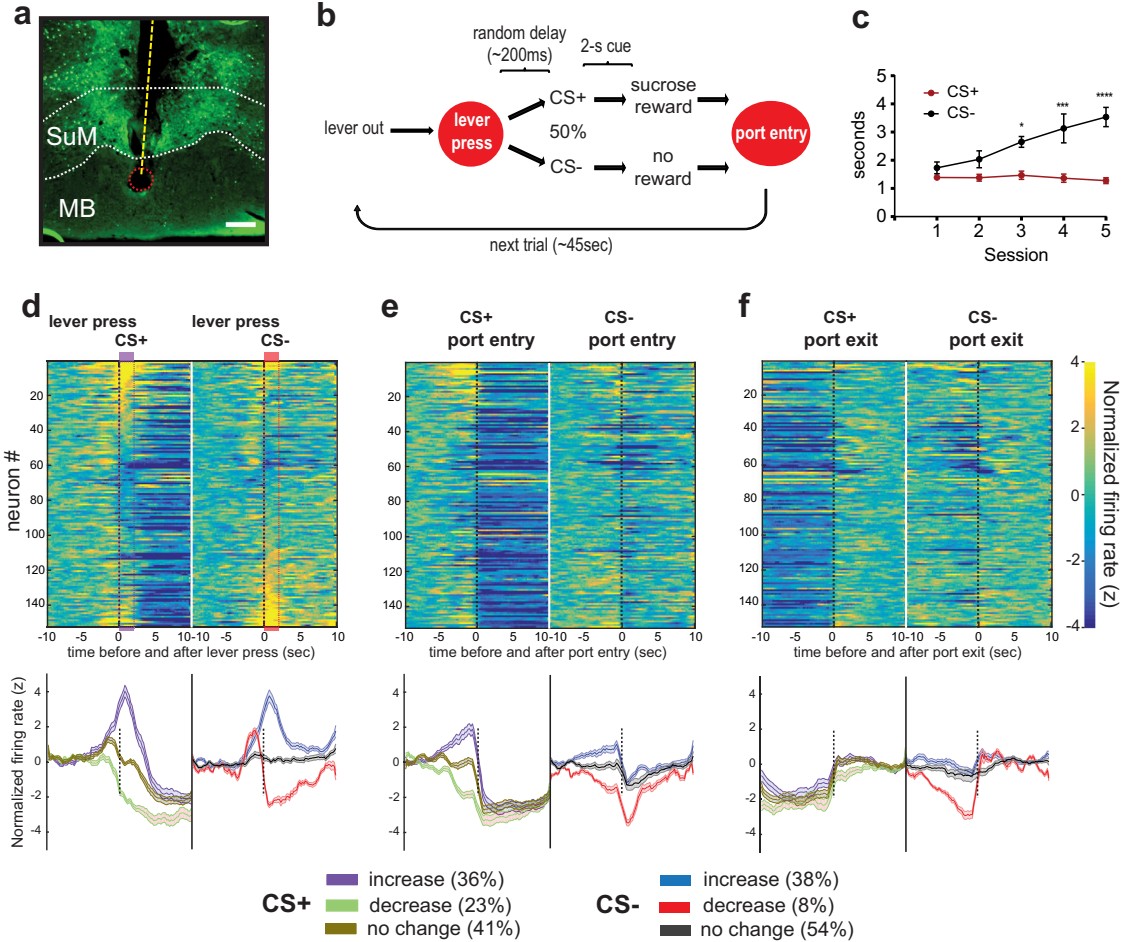

**Fig. 5 Consummatory behavior decreases the activity of SuM neurons. a** Example histology showing tetrode bundle track (dotted yellow line) passing through SuM with the tip marked by lesion (dotted red circle) ending in dorsal MB. Green fluorescence is ChR2-eYFP expression. Scale bar = 200 μm. **b** Diagram showing operant sucrose-seeking procedure with cue discrimination. **c** Latencies to enter sucrose port (mean ± SEM; $n = 6$ mice) after leaver pressing and hearing either CS+ or CS− cues during training sessions. $^*p < 0.05$, $^{***}p < 0.001$, $^{****}p < 0.0001$, Bonferroni test. **d–f** Top: Heatmaps of normalized firing rates of SuM neurons during lever presses that initiate CS+ or CS− events (**d**), nose poke entry (**e**), and nose poke withdrawing (**f**) that follow CS+ or CS− lever presses. Neurons are arranged in the same sequence from top to bottom across all heatmaps. Bottom: Traces indicate mean (±SEM) normalized firing rate of neurons that are categorized as increasing, decreasing, or non-responding to CS+ and CS− with the same time scale as the raster plot above. The percentage of neurons in each category is shown in the boxes below. Source data are provided as a Source Data file.

(Supplementary Fig. 6c). Most neurons that significantly correlated to locomotor activity did so with relatively small *R*-values (Supplementary Fig. 6d). In addition, we examined SuM neuron activity with respect to grooming as mice occasionally performed this behavior during the session. We found that while subsets of SuM neurons show changes in firing rate either before or after initiating or ending a grooming bout, the majority show no change in firing rate to these behavioral events (Supplementary Fig. 6e). These results suggest that locomotor activity or grooming is not strongly correlated with unit activity of SuM neurons, and therefore the decreased activity of SuM neurons is attributable to consummatory behavior, rather than behavioral immobility associated with the reward consumption.

With respect to the reward-seeking task, we found that SuM neurons responded heterogeneously. During the 2 s before lever pressing, some of the SuM neurons increased (38.8%, 59/152) and 8.6% (13/152) decreased firing rates, while others remain unchanged (52.6%, 80/152). The presentation of CS+ resulted in increased firing rates for 36.2% ($n = 55/152$) of neurons, decreased for 23.0% ($n = 35/152$), and no change for 40.8% ($n = 62/152$) (determined by *z*-score normalized to time period −10 to −6 s before lever press events with values ≥1, ≤1, or absolute

value <1, respectively). The presentation of CS− resulted in 38.1% increase ($n = 58/152$), 7.9% decrease ($n = 12/152$), and 53.9% no change ($n = 82/152$) in firing rate (Fig. 5d–f). These activity patterns of SuM neurons appear to be similar whether they project to the septum or not (Supplementary Fig. 7). In summary, while SuM neurons did not selectively respond for reward-associated events, they tended to increase activity upon environmental events. Together with their decreased activity during consummatory behavior, we propose that SuM neurons are important for exploration during appetitive tasks.

**Inhibition of SuM neurons disrupts sucrose-seeking, but not consummatory responses.** Our in vivo single-unit recording data suggest that SuM neuron activity is involved in behavioral interaction with the environment during reward seeking, and less so during consummatory behaviors. We examined whether the inhibition of SuM neurons selectively disrupts behavioral interaction with the environment, namely approach responses, while having little or no effects on consummatory responses. Mice were trained to perform the same sucrose-seeking task described above (Fig. 6a), and then several metrics of approach and consummatory responses for sucrose were assessed while temporarily

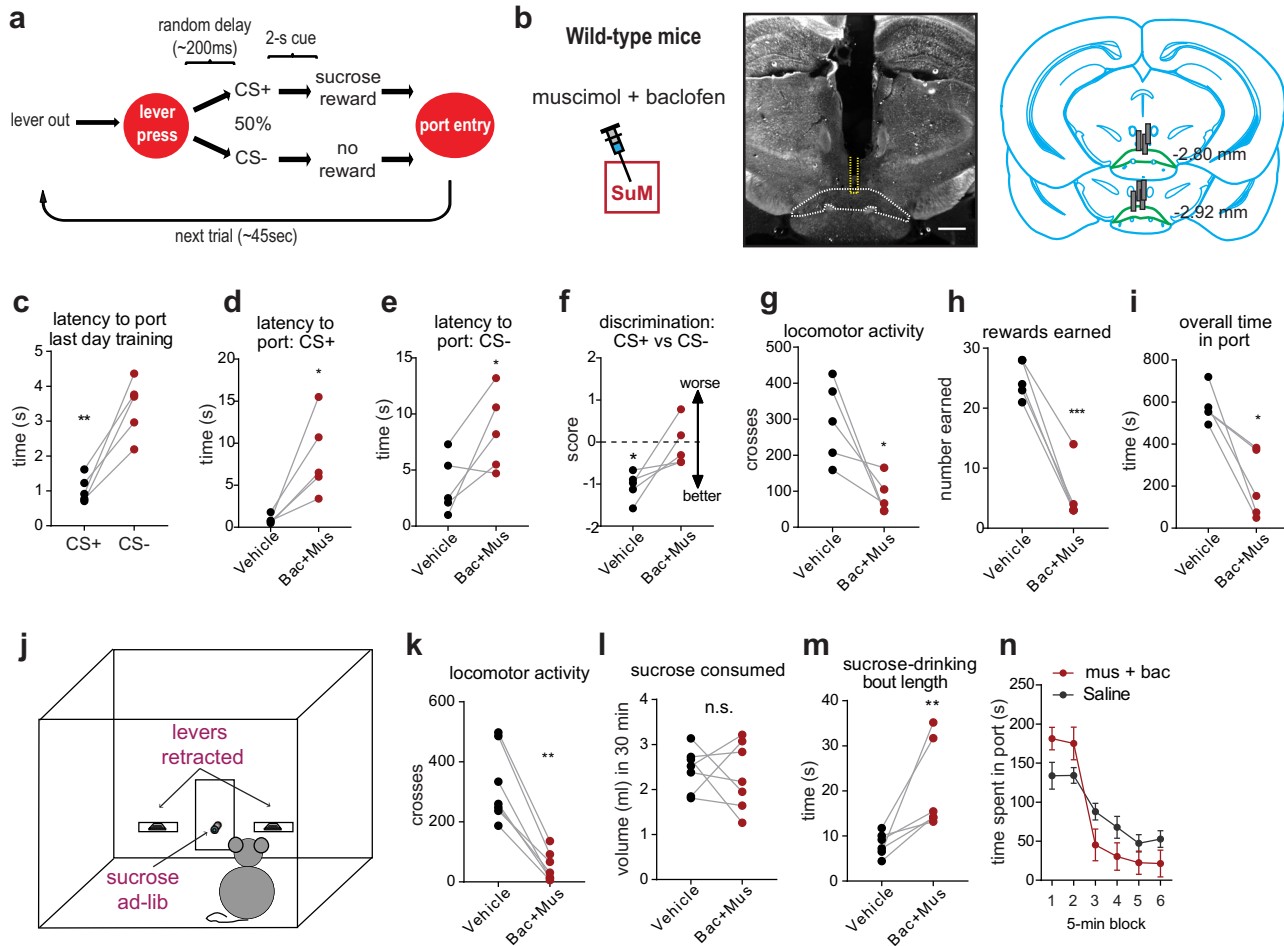

**Fig. 6 Inhibition of SuM neurons disrupts reward seeking, but not consumption. a** Diagram showing operant sucrose-seeking procedure with cue discrimination. **b** SuM cannulation for muscimol + baclofen infusions. Left: Schematic showing a guide cannula targeting the SuM. Middle: Darkfield histological example showing SuM guide-cannula track with injection-cannula track outlined in dotted yellow line; scale bar = 500 μm. Right: Diagram showing coronal sections with the placements of cannulae. **c–i** Performance measures during the operant sucrose-seeking task. $^*p < 0.05$, $^{**}p < 0.01$, $^{***}p < 0.001$, paired $t$-test ($n = 5$ mice). **j** Schematic of mouse operant-conditioning chamber with two levers retracted. **k–m** Performance measures during the sucrose drinking test. $^{**}p < 0.01$, paired $t$-test ($n = 7$ mice). **n** Here, 5 min sucrose intake over the course of the 30 min test are shown (mean ± SEM, $n = 6$). Source data are provided as a Source Data file.

inhibiting SuM neurons by intracranial infusions of a mixture of the $GABA_A$ and $GABA_B$ receptor antagonists, muscimol and baclofen (mus+bac; Fig. 6b)[45]. All mice learned to discriminate between cues by the end of training (Fig. 6c), while microinfusions of mus+bac into the SuM increased latency to approach the sucrose port following both CS+ and CS− cues (Fig. 6d, e); caused poorer cue discrimination (Fig. 6f); and decreased general locomotor activity during the task (Fig. 6g). In addition, intra-SuM mus+bac decreased the overall number of sucrose rewards earned (Fig. 6h), thereby decreasing time spent in the sucrose port (Fig. 6i). Thus, intra-SuM infusions of mus+bac disrupted both approach behaviors and appetitive response measures associated with reward seeking. These observations could be explained by drowsiness, reduced approach responses, reduced appetite for sucrose, or one or a combination of these consequences.

To address this issue, we conducted another experiment in the same chamber with levers retracted (Fig. 6j). Ad-lib sucrose was available over a 30 min period after intra-SuM mus+bac injections in the same mice used for the above experiment. In this experiment, we replicated decreased locomotor activity (Fig. 6k) and found no significant difference in overall sucrose consumption between mus+bac and vehicle infusions (Fig. 6l). Interestingly, mus+bac injections increased the bout length of

sucrose drinking (Fig. 6m) and caused them to consume sucrose more at the beginning of the session (Fig. 6n; $6_{epoch} \times 2_{treatment}$ ANOVA; $F_{interaction}(5,25) = 4.579$, $p < 0.005$). These results are not consistent with drowsiness or reduced appetite, but with the notion that SuM inhibition disrupts interaction with the environment, including reward-approach responses and exploratory behavior, indicated by locomotor activity. Similarly, the inhibition of the VStr-VTA DA pathway is also known to disrupt approach and locomotor behavior while having little or no effect on consummatory behavior[16,19].

**SuM-to-septum GLU ICSS depends on DA transmission.** Because of the emerging functional similarities between SuM GLU neurons and VTA DA neurons described in experiments above, we examined whether behavioral reinforcement by the stimulation of SuM-to-MS GLU neurons depends on DA transmission. To this end, we compared the effect of DAR blockade on ICSS reinforced between the SuM-to-septum GLU and VTA-VStr DA pathways. We selectively expressed ChR2 in SuM GLU or VTA DA neurons to stimulate their terminals at the septum of vGlut2-Cre mice or the VStr of TH-Cre mice, respectively (Fig. 7a). Each of these groups acquired ICSS of their respective

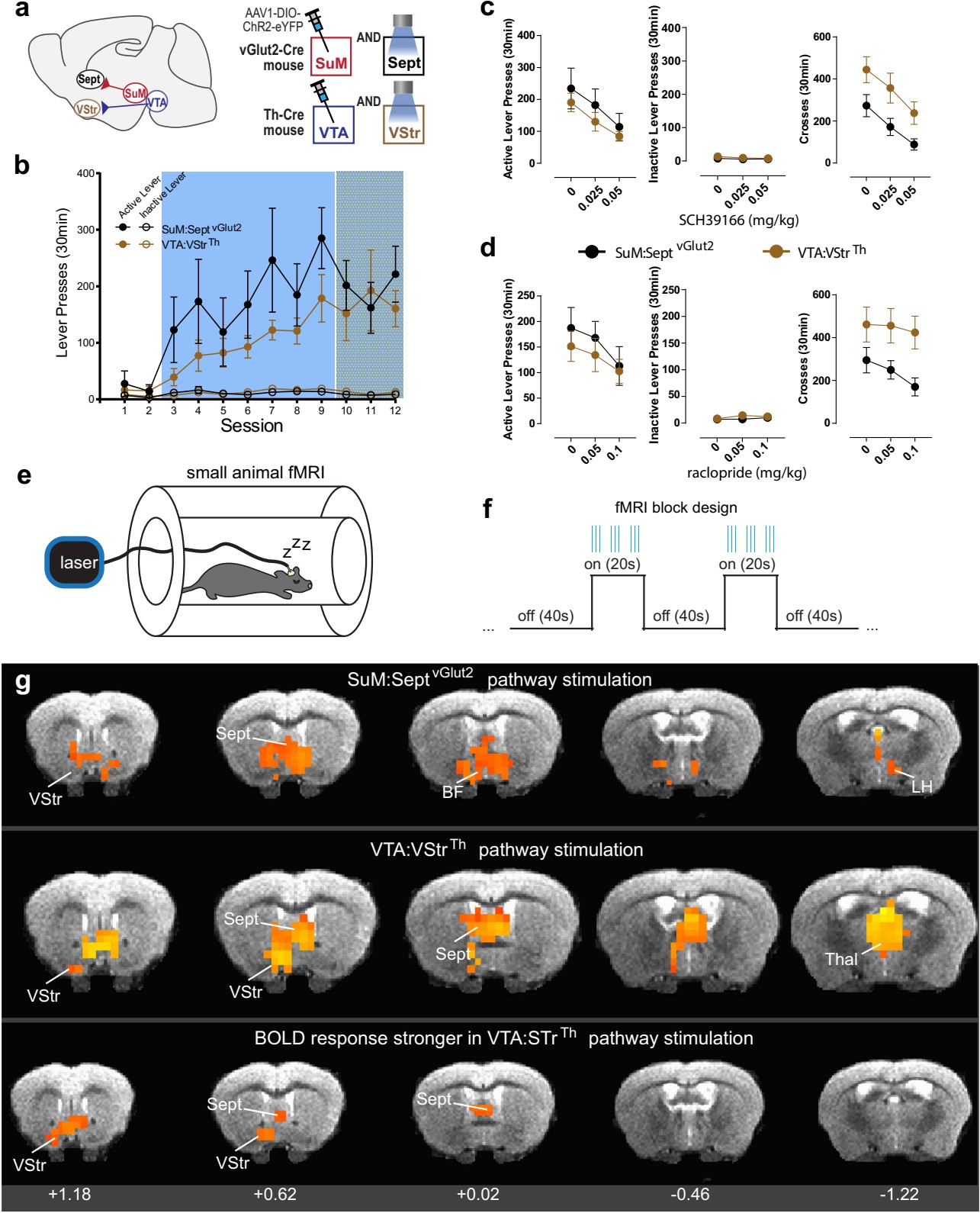

pathways (Fig. 7b, sessions 3–9). The SuM-to-MS GLU group tended to respond at faster rates than the VTA-VStr DA group. To make appropriate comparisons between these two groups, laser intensity was adjusted such that ICSS responses per session of each mouse were around 200 active presses per session (Fig. 7b, sessions 10–12). After ICSS responses of each mouse stabilized to <20% change over 2 sessions, effects of the DAR1 antagonist SCH

39166 and the DAR2 antagonist raclopride on ICSS and loco-motor activity were examined. SCH 39166 injections decreased active lever-presses (a $2_{group} \times 3_{dose}$ ANOVA: $F_{dose}(2,24) = 29.82$, $p < 0.0001$) for both SuM-to-septum GLU and VTA-VStr DA groups in a similar manner, while having no effect on inactive lever-presses ($F_{dose}(2,24) = 2.05$, $p = 0.15$) and with no interaction or group differences detected for either metric. SCH 39166

**Fig. 7 SuM-to-septum GLU ICSS depends on DA transmission and induced BOLD signal distributions are similar between SuM-to-septum GLU and VTA-VStr DA pathway stimulation. a** Schematics showing sites of AAV injections and optic fibers. **b** Lever presses per session (mean ± SEM; SuM: Sept$^{vGlut2}$ $n = 7$ mice, VTA:VStr$^{DA}$ $n = 7$ mice). In sessions 10–12, laser power was titrated for each individual mouse as to support ICSS rates ~200 active presses per session. **c, d** Effects of low and high doses of DAR1 (**c**) and DAR2 (**d**) antagonists on active and inactive lever-press rates and locomotor activity (mean ± SEM; SuM:Sept$^{vGlut2}$ $n = 6$ mice, VTA:VStr$^{DA}$ $n = 8$ mice). See text for description of statistical differences. **e** Cartoon showing an anesthetized mouse with passive stimulation in an fMRI scanner. **f** Diagram of block design for experimenter-delivered stimulation-trains during discrete ON/OFF epochs. **g** BOLD response patterns (SuM:Sept$^{vGlut2}$ $n = 7$ mice, VTA:VStr$^{DA}$ $n = 6$ mice). Coronal sections roughly correspond to Franklin and Paxinos Mouse Brain Atlas images 21, 26, 31, 35, and 41 (from left to right). Only data from diencephalic regions are shown, as the present fMRI method was not optimized for detecting lower brain structures such as the VTA. All BOLD responses shown are significant, $p_{corrected} < 0.01$, voxel-wise beta-weighted $t$-test against zero using 3dttest. Source data are provided as a Source Data file.

injections also decreased locomotor activity for both groups in a similar degree ($F_{dose}(2,24) = 39.52$, $p < 0.0001$) but the VTA-VStr DA group displayed greater locomotor activity than the SuM-to-septum GLU group during ICSS sessions ($F_{group}(1,12) = 4.79$, $p < 0.05$) (Fig. 7c). Similarly, raclopride injections significantly decreased active lever-presses ($F_{dose}(2,24) = 10.49$, $p < 0.001$) and locomotor activity (crossing $F_{dose}(2,24) = 5.36$, $p < 0.05$), but not inactive lever-presses ($F_{dose}(2,24) = 2.71$, $p < 0.09$) (Fig. 7d). Additionally, the VTA-VStr DA group again tended to display greater locomotor activity than the SuM-to-septum GLU group ($F_{group}(1,12) = 4.59$, $p = 0.053$). These results suggest that the excitation of VTA-VStr DA neurons was accompanied with stronger movement arousal than that of SuM-to-septum GLU neurons and that the reinforcing effect mediated by SuM-to-septum GLU neurons depends on DA transmission to a similar degree as that mediated by VTA-to-VStr DA neurons.

**fMRI: induced BOLD signal distributions are similar between SuM-to-septum GLU and VTA-VStr DA pathway stimulation.** We next examined whether the stimulation of SuM-to-septum GLU neurons activates brain structures similar to that of VTA-VStr DA neurons. fMRI imaging was performed in the same mice used for the DAR antagonist experiments. The mice were anesthetized and received experimenter-delivered stimulation of these two pathways (Fig. 7e, f). Similar patterns of blood-oxygen-level-dependent (BOLD) activation were observed upon stimulation between the two pathways (Fig. 7g, top and middle). The photostimulation of either pathway significantly increased BOLD activity ($p_{corrected} < 0.01$) in both VStr and septum. In addition, the activation of the SuM-to-septum GLU circuit increased BOLD in areas corresponding to ventral pallidum and pre-optic areas, the lateral hypothalamus, and midline thalamic nuclei. The activation of the VTA-VStr DA pathway additionally increased BOLD in a large portion of the anterior thalamus. Interestingly, a between-group comparison showed that VTA-to-VStr DA pathway stimulation activated both the VStr and septum more strongly than SuM-to-septum GLU circuit stimulation (Fig. 7g, bottom). These results indicate that SuM-to-septum GLU pathway stimulation can activate the VStr, while VTA-VStr DA pathway stimulation can activate the septum. Although technical limitations did not enable us to detect reliable signals in the deep brain structures, including the SuM and VTA, the results support the notion that self-administration behavior reinforced by the stimulation of the SuM-to-septum GLU pathway share common circuitry with that reinforced by the stimulation of the VTA-VStr DA pathway.

**The excitation of SuM-to-septum GLU neurons activates VTA-to-VStr DA neurons.** We conducted several experiments to examine functional linking between SuM-to-MS GLU neurons and VTA-VStr DA neurons. We performed three experiments using fiber-photometry imaging procedures in awake, freely

moving mice, and one tracer experiment. First, we found that the stimulation of SuM-to-septum GLU neurons activated VTA-DA neurons (Fig. 8a and Supplementary Figs. 8a and 9a). Second, we found that the stimulation of MS GLU neurons increased DA release in the VStr (Fig. 8b and Supplementary Figs. 8b and 9b). Third, we showed that MS GLU neurons project to the paranigral zone of the VTA and observed numerous terminal boutons of MS GLU neurons within the VTA (Fig. 8c). Finally, we found that the stimulation of MS-to-VTA GLU neurons increased DA release in the VStr (Fig. 8d and Supplementary Figs. 8c and 9c). Moreover, we gave mice the opportunity to lever press for the stimulation of MS-to-VTA GLU neurons and found that the levels of ICSS responding were positively correlated with those of VStr dLight signals (Fig. 8d). These results suggest that SuM-to-septum GLU neurons can regulate VTA-VStr DA neurons for reinforcement via MS-to-VTA GLU neurons.

**Discussion**
We found that SuM GLU neurons regulate reinforcement and motivation, and propose that such roles of SuM GLU neurons are mediated, in part, by their projections to septal GLU neurons that, in turn, project to the VTA, which regulate VTA-to-VStr DA neurons. Specifically, we propose that SuM GLU neurons regulate the host's interaction with the environment (arousal$^{interaction}$ for a shorthand). It is not clear at this time whether the same SuM GLU neurons and the same MS GLU neurons mediate both reinforcement and motivation. Although our discussion below may imply the SuM-to-MS-to-VTA pathway for reinforcement and SuM-to-MS-to-hippocampal formation for arousal$^{interaction}$, such characterization is most likely too simplistic, and this issue should be addressed by future research.

Many lines of research suggest the role of SuM neurons, including those projecting to the MS, and MS neurons in arousal. Historically, multiple neural populations have been implicated in arousal, including norepinephrine, histamine, and cholinergic and orexinergic systems, which project broadly to the forebrain, and each population must be involved in different aspects or types of arousal[46,47]. Briefly, the SuM and the MS are implicated in arousal with the following findings: First, SuM GLU neurons are shown to regulate wakefulness; the stimulation of SuM GLU neurons prolongs wakefulness, while the inhibition does not necessarily cause somnolence[9]. Second, our unit activity experiment showed that the activity of SuM neurons markedly and uniformly decreased when the mice consumed sucrose reward. Third, the inhibition of SuM neurons disrupted sucrose-seeking responses and decreased locomotor activity without decreasing sucrose-consummatory responses. Fourth, disinhibition of SuM neurons induced by GABA$_A$ receptor blockade markedly increases locomotor activity and results in robust c-Fos expression in the SuM as well as in the septum[8]. Fifth, intra-septal injections of AMPA increased behavioral interaction with an unconditioned VS. Sixth, neural activity of septal GLU neurons is positively correlated with locomotor speed, accompanied by

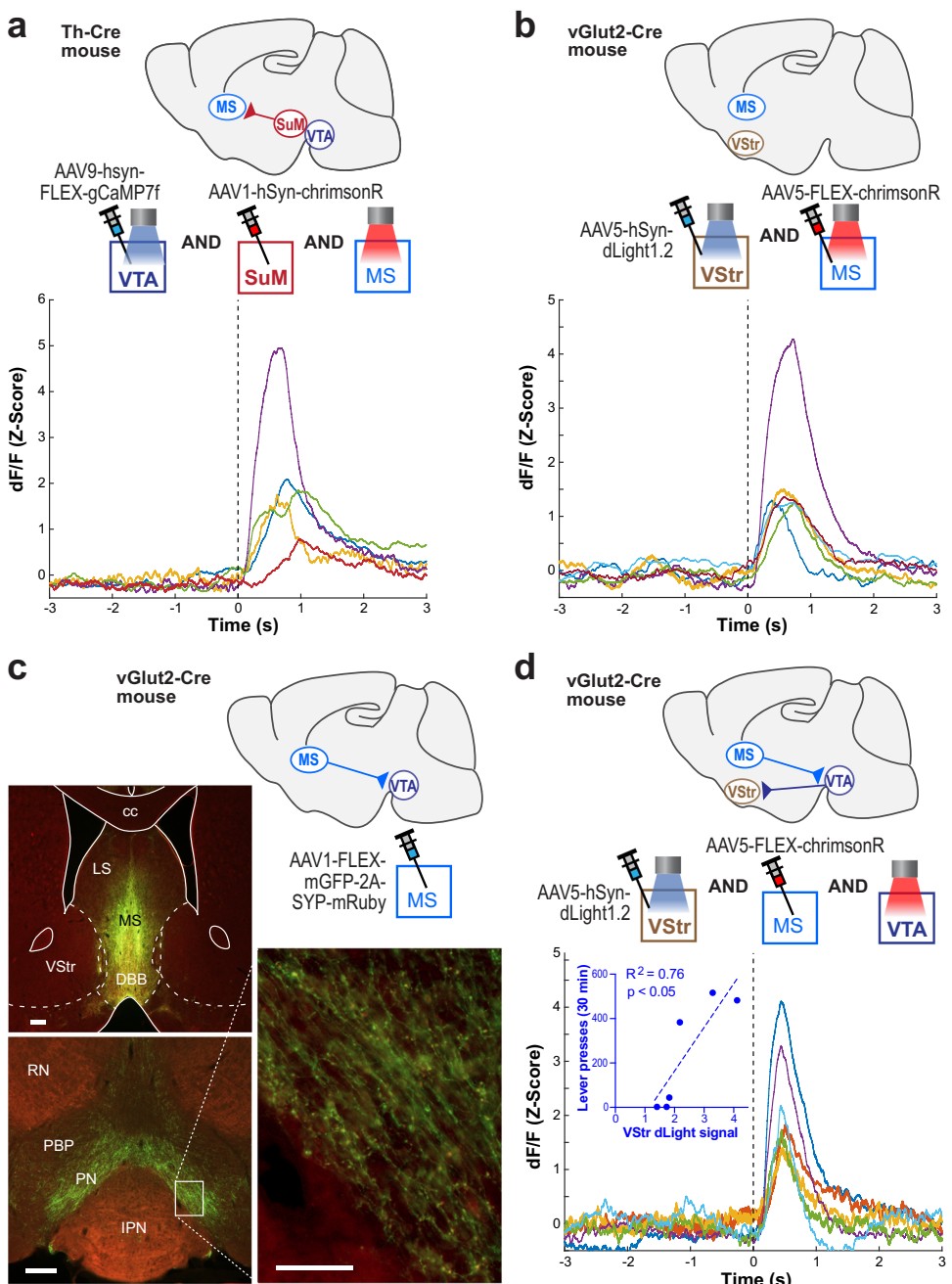

**Fig. 8 The excitation of SuM-to-septum GLU neurons activates VTA DA neurons and vice versa. a** Top: Schematics showing sites of AAV injections and optic fibers. Bottom: Mean traces of GCaMP transients evoked by the stimulation of SuM-to-MS GLU neurons for individual mice ($n = 5$). **b** Top: Schematics showing sites of AAV injections and optic fibers. Bottom: Mean traces of dLight transients evoked by the stimulation of MS GLU neurons for individual mice ($n = 6$). **c** Fluorophores induced by MS injection of AAV-FLEX-mGFP-2A-SYP-mRuby in Vglut2-Cre mice ($n = 3$). This vector fills the entire Cre (i.e., GLU)-containing cells with mGFP and their terminal boutons with mRuby. Scale bars, left two = 200 μm; scale bar, right = 50 μm. Top left: Example photomicrogram showing injection site. Bottom left: Example photomicrogram showing fluorophores expressed in the VTA. Right: Enlarged squared region showing terminal boutons in the VTA. Abbreviations: cc, corpus callosum; DBB, diagonal band of Broca; LS, lateral septum; MS, medial septum; PBP, parabrachial pigmented nucleus of the VTA; PN, paranigral nucleus of the VTA; RN, red nucleus; VStr, ventral striatum. **d** Top: Schematics showing sites of AAV injections and optic fibers. Bottom: Mean traces of dLight transients evoked by the stimulation of MS-to-VTA GLU neurons for individual mice ($n = 6$). Inset: Correlation coefficient between evoked dLight signals in the VStr and the ICSS levels in session 5.

hippocampal theta oscillations[42,48]. These findings suggest roles of the SuM and the MS in arousal[interaction].

The supramammillo-septal system has been implicated in arousal related to cognitive processes. Previous research found that SuM neurons modulate theta oscillations through the projections to the MS, which in turn regulates theta oscillations[2,3]. This mechanism can explain how theta oscillations are active during environmental

interaction, but not during self-care activities such as food consumption[44], an observation that is similar to those of SuM neuron activity. Therefore, SuM neurons most likely play an important role in spatial navigation and latent learning, i.e., learning without rewards, involving spatial relationship of landmarks or events involving objects and other experiences, which are hallmark learning processes associated with theta oscillations[49].

The arousal role of SuM GLU neurons has little to do with the anticipation of rewards, but more to do with coordinating activities of other brain structures with respect to space navigation. While some SuM neurons increased firing rates just prior to lever pressing, these activities do not seem to reflect prediction since the presentation of CS+ or CS− resulted in variable responses: increase, decrease, or no change, and unit activity did not discriminate for a large part between CS+ and CS−. However, SuM neurons are involved in coordinating other structures for navigating the host to rewards[50]. The inhibition of SuM neurons reduces coherences between theta oscillations and both prefrontal cortical and reuniens thalamic activity. Interestingly, while unit activities of the prefrontal cortex and the reuniens contain trajectory information for rewards, SuM activity does not[50]. Thus, SuM neurons appear to coordinate activities of other structures in environmental interaction without guiding behavior. Such SuM's capacity in coordinating hippocampus-related structures[3] is consistent with its proposed role in arousal[interaction].

An interesting implication is that the supramammillo-septal system plays an important role in latent learning and motivated behavior in the absence of rewards. As discussed above, SuM GLU neurons responds to broad environmental stimuli, including positive and negative, conditioned and unconditioned stimuli, and they are important in regulating the host's interaction with the environment, but not food intake. Animals and humans voluntarily engage in behavioral activities that are not readily linked with canonical rewards, such as wheel-running (in rodents), dancing (in humans), and exploration (animals and humans), which can be reinforcing. Animal research shows that animals work to obtain the opportunity for exploration[51,52] or wheel-running[53,54], and these activities results in significant c-Fos expressions in the SuM[55,56]. Therefore, in contrast to DA neurons, which play an important role in motivation and learning involved in canonical rewards, supramammillo-septal neurons are important in motivation and learning in the absence of canonical rewards.

Additional notable implications concern addiction, depression, and the regulation of DA neurons. While nicotine is found to act at the SuM for its reinforcing effect[6], the present data implicate the supramammillo-septal system in substance abuse more prominently than previously thought. The present study found that MS GLU neurons specifically MS-to-VTA GLU neurons support optogenetic self-stimulation and regulate VTA DA neurons and VStr DA levels. In addition, fMRI data indicated that VTA-to-VStr DA neurons can regulate SuM-to-septal GLU neurons. These results raise a possible positive-feedback relationship between the systems and raises the possibility that any substance that stimulates VTA DA neurons also activates SuM-to-septal GLU neurons and, in turn, further activates the VTA DA neurons. This new understanding of functional connections between these systems has important implications for future research on neural mechanism of substance abuse, and also normal motivated behaviors and psychiatric disorders such as depression.

Finally, brief discussion is warranted on the finding that the stimulation of SuM GLU neurons projecting to the PVT is aversive. This unexpected finding is interesting and potentially important because recent findings implicated the PVT in aversive-related affective functions. Opioid withdrawal is found to induce aversion via the activation of PVT neurons projecting to the VStr[30]. Similarly, PVT neurons projecting to the central amygdaloid nucleus are shown to regulate the acquisition and expression of fear memory[32]. Thus, it is possible that SuM neurons participate in such functions. Consistent are the findings that morphine withdrawal induces high c-Fos expression in the SuM, and the presentation of conditioned stimuli associated with sickness induces expressions of c-Fos in the SuM and PVT[57]. Our results suggest that SuM GLU neurons provide an afferent pathway to PVT for aversion. The notion of a role of

**Table 1 List of AAVs.**

| AAV | Titer (vg/m) | Source |
|---|---|---|
| AAV1-hSyn-ChR2(H134R)-EYFP | $7 \times 10^{12}$ | GEVVC |
| AAV1-Ef1a-DIO-hChR2(H134R)-EYFP | $7 \times 10^{12}$ | Addgene |
| AAV1-Syn-ChrimsonR-tdTomato-WPRE | $2.3 \times 10^{13}$ | Addgene |
| AAV2-retro-EF1a-Nuc-flox(mCherry)-eGFP | $3.4 \times 10^{12}$ | GEVVC |
| AAV9-Syn-FLEX-GCaMP7f-WPRE | $2.7 \times 10^{11}$ | Addgene |
| AAV5-Syn-FLEX-rc[ChrimsonR-tdTomato] | $2.0 \times 10^{13}$ | Addgene |
| AAV5-hSyn-dLight1.2 | $8.7 \times 10^{12}$ | Addgene |
| AAV1-syn1-FLEX-mGFP-2A-SYP-mRuby | $6.9 \times 10^{11}$ | GEVVC |
| AAV1-EF1a-DIO-Nuc-dTom | $7.2 \times 10^{12}$ | GEVVC |

SuM in arousal[interaction] is consistent with an involvement in positive and negative emotional valence, since a balance between these states is necessary for appropriate environmental interaction. However, further research is needed to understand how SuM neurons participate in aversion and whether or how SuM neurons involved in negative affective state are independent of those involved in positive state.

## Methods

### Animals
*Mice.* We used adult male and female mice (2–4 mo.) weighing 25–35 g at the time of surgery. C57BL6J were obtained from Jackson Labs, while transgenic mice were obtained and then bred at the National Institute on Drug Abuse Intramural Research Program animal facility: (Vglut2)::IRES-Cre (Slc17a6tm2(cre)Lowl); Jackson Labs), (vGat)::IRES-Cre (Slc32a1tm2(cre)Lowl); Jackson Labs), ChAT-IRES-Cre::frt-neo-frt (Chattm2(cre)Lowl); Jackson Labs), and (Th)::IRES-Cre knock-in mice (Lindeberg et al.[58]). When not being tested, all mice were individually housed in a vivarium and maintained at consistent temperature (70–74 °F) and humidity (35–55%) on a 12:12 h light–dark cycle (lights on at 07:00 a.m.).

*Rats.* We used 111 male Wistar rats (Envigo, Indianapolis, IN) weighing 250–350 g at the time of surgery. The colony room was maintained at consistent temperature and humidity on a reverse 12 h dark–12 h light cycle (light on at 8:00 p.m.).

For all animals performing intracranial self-stimulation or self-administration, food and water were freely available except during testing. Mice performing sucrose-seeking were water restricted for ~20 h before training and were provided additional access to water in home cages in the afternoon before returning to the animal colony. All procedures were approved by the Animal Care and Use Committee of the Intramural Research Program, National Institute on Drug Abuse and were in accordance with the Guide for the care and use of laboratory animals (National Research Council[59]).

### Mouse in vivo optogenetic stimulation
*Viral vectors.* AAVs (Table 1) were obtained from NIDA Genetic Engineering and Viral Vector Core (GEVVC; Baltimore, MD), and Addgene (Watertown, MA).

*Surgery.* Coordinates used for various brain regions are described in Table 2. These coordinates represent the placement of the tip of optic fiber. Injections of ChR2 viral vectors are always 0.2 mm ventral to these coordinates. Mice were anesthetized using isoflourane and placed in a stereotaxic apparatus for surgery. For each experiment, one of the viral vectors (Fig. 1a–e: 300 nL; Fig. 1f–h: 150–200 nL, and 50 nL in a subset of vGlut2-Cre mice; Fig. 2: 200 nL; all other viral injections for other experiments were 150 nL) was microinjected by a syringe pump (Micro 4, World Precision Instruments) at 50 nL/min, with additional 5–10 min waiting before removal of the injection needle (34 gauge, beveled). Then, optic fibers (200 μm core size with numerical aperture of 0.37), constructed as described before[60], were chronically implanted secured on the skull with dental cement (Geristor A and B cement, Denmat; part #s 4506 and #034522101). For experiments stimulating SuM or adjacent cell bodies, hSyn-driven ChR2 was injected into the SuM, and a single optic fiber was implanted to stimulate one of the following regions: SuM, MB, or VTA. Similarly, to study SuM neuron subtypes, a Cre-dependent ChR2 was injected into SuM and fibers placed above SuM. For experiments studying SuM neuron terminals, Cre-dependent ChR2 was injected into SuM and fibers placed in the septum (either MS or LS) and PVT or vSub. In these studies, one cohort received implants in both the vSub and septal area; another cohort received implants in both PVT and septal area; a third cohort had a single implant in the ventral septal area/DBB, to stimulate ventrally to those placed in the dorsal septal area; and a fourth cohort received injections in the VTA and optic fibers in the septum. For real-time place preference experiments studying septum and PVT, each mouse received injections of Cre-dependent ChR2 in SuM and one fiber in septal area and another in PVT. For experiments directly

**Table 2 Stereotaxic coordinates for injections and implantations.**

| Region | AP | ML | DV | Deg |
|---|---|---|---|---|
| Mouse | | | | |
| SuM | −2.7 | 0.0 | −4.6 | 0° |
| MB | −2.7 | 0.0 | −5.1 | 0° |
| VTA | −3.2 | 0.5 | −4.0 | 0° |
| MS | +0.8 | 0.0 | −3.5 | 0° |
| LS | +0.8 | 0.2 | −3.1 | 0° |
| DBB | +0.8 | 0.2 | −3.8 | 0° |
| vSub | −3.3 | 2.8 | −4.3 | 0° |
| PVT | −1.8 | 0.0 | −2.7 | 0° |
| VStr | +1.0 | 0.5 | −4.0 | 0° |
| Rat | | | | |
| MS | +0.6 to +1.1 | 0.0 to +1.0 | −4.1 to −5.9 | 0° |
| LS | +1.1 to +1.4 | +1.1 | −5.5 to −6.0 | 20° |
| DBB | +1.3 | +1.9 | −7.0 to −7.6 | 20° |
| VStr | +1.5 to +2.0 | +1.5 to +1.6 | −6.8 to −7.6 | 20° |

AP = Anterior/posterior. ML = Medial/lateral. DV = Dorsal/ventral. Deg = degree of angle from vertical. All DV coordinates are measured from skull surface. All distances in mm from bregma. Negative AP indicates positions posterior to bregma.

stimulating septal neurons, hSyn-driven or Cre-dependent viral vector injection and optic fiber placements were in the LS or MS.

*Recovery and habituation.* For cell-body stimulation a 2 week recovery/incubation period was followed by handling for 3 days, 5–10 min each day, by the experimenter conducting behavioral tests, followed by 30 min habituation session in the test chambers the day before experimentation began. The same post-surgery procedure was followed for neuron terminal stimulation experiments, except a 6 week recovery/incubation period was given to allow for trafficking of opsins to projection terminals.

*Optogenetic intracranial self-stimulation.* Self-stimulation experiments were conducted in standard operant-conditioning chambers (15.9 × 14.0 × 12.7 cm; Med Associates, St. Albans, VT) equipped with two levers and a small cue lamp above each lever, a house lamp, and 4 pairs of infrared detectors along the cage floor to detect locomotion. Mice were gently connected to a patch cable connected to a 473 nm laser for ChR2 stimulation via an optical swivel. Computer software (MEDPC; Med Associates, St. Albans, VT) controlled a pulse generator (MASTER 9; AMPI, Jerusalem, Israel) that controlled lasers. Each ICSS session lasted for 30 min, and sessions were typically separated by 1 day. For the first 10 sessions, a response on the 'active' lever illuminated a cue lamp above the lever for 1 s. This cue procedure has been commonly used in intravenous self-administration procedure to facilitate the acquisition of lever pressing. A response on the 'inactive' had no programmed consequence. To confirm that mice display ICSS without the visual cue, no cue was provided in session 11 and subsequent sessions.

During the first 2 sessions, lasers were turned off and intracranial photostimulation was not delivered to assess baseline response levels. During sessions 3–7, active lever pressing was rewarded with an intracranial photostimulation via the implanted optic fiber. For all optogenetic ICSS procedures in this study, we maintained the following parameters for delivery of photostimulation: 15 pulses of blue light (473 nm), with 5 ms pulse duration delivered at 25 Hz. For experiments targeting SuM cell bodies, we used 5 mW light intensity at the tip of the optic fiber implant in the brain to limit the spread of light to adjacent anatomical structures. In experiments stimulating terminals, as well as to stimulate septal neuron cell bodies, we used 10 mW laser intensity at the tip of the optic fiber in the brain to better recruit these larger populations of ChR2-expressing tissue. Acquisition sessions 3–7 were followed by 3 extinction sessions where active lever pressing delivered no photostimulation (sessions 8–10). Thereafter, 2 reacquisition sessions were carried out with photostimulation (sessions 11–12). In addition, the mice received a lever reversal test over 4 sessions in which the assignment of active and inactive levers with respect to the right and left levers was reversed without any cue before session 15 (sessions 13–16). In other experiments, mice were tested through session 7 to test for acquisition of ICSS only. For experiments testing SuM-to-septum/PVT/vSub pathway stimulation, mice were connected to one of their fiber implants and tested in the same manner as described above through session 7. Then for sessions 8–12, testing identical to sessions 3–7 was repeated, but mice were now connected to the fiber implant in the second region. We tested the mice in a counterbalanced way such that, in sessions 1–7 half of the mice performed ICSS for one region followed by ICSS for the second region in sessions 8–12, while the other half of mice performed ICSS for the same regions in the opposite order. Before euthanasia, a subset of these mice (vSub, n = 10; PVT, n = 6; septal area, n = 6) received non-contingent photostimulation (one pulse train every second) for 15 min in operant chambers to

stimulate ChR2 in terminals and analyze c-Fos or were handled and placed similarly in chambers but received no photostimulation (n = 8) to serve as control. Mice were left in their home cage for 45 min after this final session and then euthanized. In addition, for vSub c-Fos counts, tissue ipsilateral to stimulation was compared to the same tissue contralateral to stimulation. For all optogenetic ICSS experiments, any mouse that failed to press active lever more than 15 times in total during acquisition sessions were removed from the study for lack of exposure to effects of the stimulation. When comparing ICSS rates between SuM-to-septum GLU and VTA-to-VStr DA pathway stimulation, we set lasers to achieve 10 mW laser intensity at the tip of the optic fiber in the brain during acquisition sessions and then in sessions 10–12, we titrated the laser intensity to tune lever pressing rates in each group to roughly 200 lever presses in the 30 min session. To achieve this, in general, laser power was increased up to 20–30 mW for most VTA-to-VStr DA mice and was decreased down to 3 mW for some SuM-to-septum GLU mice. One SuM-to-septum GLU mouse failed to reach consistency in lever pressing, so was excluded from DA antagonist experiments. One VTA-to-VStr DA did not acquire lever pressing behavior until later sessions (sessions 10–14) and was excluded from acquisition analysis but included in DA antagonist experiments.

Note that in our experiment testing SuM GLU/GABA/DAergic cell bodies for ICSS, 4 mice that failed to lever press more than 15 times total throughout the 7 day training period were removed from the study. Four vGat-cre mice had optic fibers placed posterior to SuM and were removed from the study. Three TH-cre mice did not show expression of ChR2-eYFP, likely due to an injection error, and were removed from the study. We note that two TH-Cre mice showed robust lever pressing rates, far higher than any other self-stimulating mouse in this study, and akin to optogenetic stimulation of midbrain DA neurons previously observed by our group[21]. Indeed, these mice had strong viral expression throughout the extent of the VTA, an observation consistent with the view that photostimulation activated VTA DA neurons; thus, they were also removed from the study.

*Real-time place preference/aversion.* Mice were acclimated in the room 1 h prior to experiments. On the first day of testing, each mouse was gently attached via either septum or PVT targeting optic fiber to a patch cable connected to a 473 nm laser for ChR2 stimulation via an optical swivel allowing for freedom to explore the entire arena (ANY-maze mouse place preference box; Stoelting Co., Wood Dale, IL). Video tracking software (EthoVision XT, Noldus, Leesburg, VA) tracked animal position and generated controlled laser via connection to a pulse generator (OPTOG-8, Doric Lenses, Quebec, Canada). After a 20 min baseline period, light pulses (15 pulses, 25 Hz, 10 mW; generated every second) were delivered only when mice centers crossed the border from center connector region of the chamber into room with striped walls. Mice were tested in this procedure on separate days, for a total of 2 sessions, to test the effects of each pathway in a counterbalanced manner.

*DA antagonist procedures.* DAR antagonists were made to stock concentrations and frozen. In the morning of drug administration sessions, aliquots of drug were thawed and diluted to working concentrations. Drugs were diluted with sterile saline, and HCL was used to aid in dissolving as needed, after which NaOH was used to return solution to near pH 7.3. Vehicle consisted of the same amount of acid and base added as drug solutions. DAR1 antagonists R(+)-SCH-23390 hydrochloride (Sigma-Aldrich; D054); SCH 39166 hydrobromide (Tocris Bioscience; Cat. No. 2299), and DAR2 antagonist S(−)-Raclopride (+)-tartrate salt (Sigma-Aldrich; R121) were used. Mice received intraperitoneal (i.p.) injections of saline for 3 sessions (sessions 10–12) to habituate to the injection procedure. These mice were then required to exhibit stabilized ICSS rates, as determined by ≤20% change in active lever-presses over 2 sessions, before moving on to DAR1 antagonist injection sessions. Mice were tested in 3 consecutive daily sessions, i.e., 1 session per day in the following order: Receiving in the first session vehicle, followed by a low dose of DAR1 antagonist SCH36199 (0.025 mg/kg) on the next day, and then a high dose of SCH36199 (0.050 mg/kg) on the third session. Next, mice performed 4–10 sessions of free ICSS before being tested with the DAR2 antagonist, raclopride, in the same daily session dose scheme—vehicle, followed by low dose (0.05 mg/kg), and lastly the high dose (0.1 mg/kg).

*Fiber photometry for calcium imaging with optogenetic stimulation.* We used the fiber-photometry system manufactured by Doric Lenses (Quebec, Canada). The light pulse generator, consisting of a driver and LED units, produced light wavelengths (465 and 405 nm) in sinusoidal waveforms (208 and 530 Hz, respectively), which were fed into a fluorescence mini-cube via patchcables (NA: 0.48; core diameter: 400 μm). The mini-cube combined these wavelength lights and sent the combined beam into the brain via a patchcable (NA: 0.48; core diameter: 400 μm) connected to the implanted optic fiber (NA: 0.48; core diameter: 200 or 400 μm). The same patchcable/optic fiber assembly, in turn, carried the emission of GCaMP or dLight (525 nm) as well as control (430 nm) back to the mini-cube, which then separated emission bandpass with beamsplitters and sent them to photoreceiver modules (Newport: model 2151) via patchcables (NA: 0.48; core diameter: 600 μm). The photoreceiver modules quantified signals and sent them to the Fiber Photometry Console, which was controlled by Doric Neuroscience Studio software and synchronized the acquisition of the data with the output of lights. Recording data were collected at the rate of 1200 Hz.

Med Associates' system (Fairfax, VT) produced the trains (2, 4, 8, or 16 pulses at 25 Hz or 16 pulses at 25 or 50 Hz) delivered in a random order on a variable-

interval schedule with the mean interval of 15 s by controlling the pulse generator (Doric Lenses) that, in turn, controlled a laser for generating a 3 ms pulse of a 635 nm wavelength. Each mouse received 41 trains for each pulse at the MS, and signals were detected at the VTA for GCaMP and the VStr for dLight.

*ICSS following the fiber-photometry experiment.* The mice that were tested for the effects of the stimulation of MS-to-VTA GLU neurons on VStr dLight signal were placed in operant-conditioning chambers and given the opportunity stimulate MS-to-VTA GLU neurons upon a response on the active lever. Each mouse was placed in a chamber for 30 min per day for 5 days. Their active lever-presses in the last session were used for the correlation analysis with their dLight signals.

*Fiber-photometry data analysis.* We used custom-written MATLAB code to transform and analyze photometry data. We first binned the separate photometry (GCaMP or dLight) and control channels into 1 min epochs. We transformed the control channels of each bin to a linear fit of its respective raw photometry channel, and calculated dF/F for each bin with the formula (rawPhotometry -FittedControl)/FittedControl[61]. We then extracted the dF/F trace corresponding to 3 s before and after every light train, extracted the corresponding area under the curve, and performed a repeated measures ANOVA with time (before and after stimulation train) and stimulation (2, 4, 8, and 16 pulses or 25 and 50 Hz) (GraphPad Prism) on the extracted area under the curve for each region of individual animal. We performed post hoc *t*-tests with Benjamini and Hochberg correction on time for each pulse when the interaction was significant.

*Histology.* After completion of behavioral experiments, all animals were intracardially perfused with ice-cold 0.9% saline followed by 4% paraformaldehyde. Brains were coronally sectioned at 40 μm, mounted directly onto slides, and coverslipped with DAPI nuclear counterstain (#H-1200, Vector Labs, Burlingame, CA) in Mowiol 4–88 (Sigma-Aldrich, #81381). Optical fiber placements and EYFP expression were determined with fluorescent microscopy.

*c-Fos staining and counting.* Brains were coronally sectioned at 40 μm. Sections were collected alternating between 4 different wells containing 0.1 M PB buffer, then transferred to a cryoprotectant solution and kept at −80 °C until further processing. EYPF and c-Fos were labeled via immunohistochemistry using rabbit anti-c-Fos (1:2000; #SC-52, Santa Cruz, Dallas, TX), goat anti-GFP/YFP (1:3000; #A11120, Life Technologies, Grand Island, NY) with donkey anti-goat Alexa Flour 488 and donkey anti-rabbit Alexa Flour 594 (1:300; Life Technologies). Mounted sections were coverslipped with a mixture of Mowiol 4–88 and DAPI nuclear counterstain. c-Fos-ir cells were counted using ImageJ (http://imagej.nih.gov/ij/) software, by automatically counting cells within a $500 \times 500$ μM$^2$ area beneath the tip of the optic fiber in the region of interest.

### Intracranial self-administration of AMPA in rats
*Surgery.* Each rat was stereotaxically implanted with a permanent unilateral guide cannula (24 gauge) under sodium pentobarbital (31 mg/kg, i.p.) and chloral hydrate (142 mg/kg, i.p.) anesthesia. The incisor bar was set at 3.3 mm below the interaural line. Stereotaxic coordinates are shown in Table 2. Each cannula was subsequently anchored to the skull by 4 stainless-steel screws and dental acrylic, and a stainless-steel wire (31 gauge) was inserted to keep it patent. Rats were housed singly to prevent other rats from chewing the implant after the surgery, which was followed by a minimum of 5 days recovery before the start of experimentation.

*Chemicals.* (S)-AMPA (Sigma-Aldrich, MO) and ZK 200775, a competitive AMPA receptor antagonist (Tocris Bioscience, MO) were dissolved in artificial cerebrospinal fluid (aCSF) consisting of (in mM): 148 NaCl, 2.7 KCl, 1.2 CaCl$_2$, and 0.85 MgCl$_2$, pH adjusted to 7.0–7.5.

*AMPA self-administration.* Each rat was placed in an operant-conditioning chamber ($30 \times 22 \times 24$ cm; Med Associates, St. Albans, VT) equipped with a lever, a tone speaker, a cue light, and a house light. An injection cannula (31 gauge) was inserted into the guide cannula and extended 1 mm beyond the tip of the guide with the exception of experiments examining multiple injection depths, described below. Then, the guide was connected by polyethylene tubing to a micropump consisting of a drug reservoir and step motor[34] that hung a few millimeters above the rat's head. When activated, the micropump's step motor shaft turned in eight incremental steps (9° per step) over 5 s, driving the threaded shaft into the drug reservoir and, in turn, pushing a 100 nL volume out of the reservoir into the brain. Each session lasted 90 min or until the rats received a total of 60 infusions.

A response on the lever delivered an infusion over 5 s, during which a tone and cue light above the lever were presented, followed by a 5 s timeout period during which responding on the lever produced no infusion. The majority of rats ($n = 67$) received infusions of AMPA concentrations as follows: 0, 0.01, 0.01, 0.05, 0.05, 0.25, and 0.25 mM in this order over 7 sessions. Other rats received AMPA infusions as follows: 0, 0.1, 0.1, 0.25 and 0.25 mM over 5 sessions ($n = 16$); 0, 0.03, 0.03, 0.1, 0.1, 0.3, and 0.3 mM over 7 sessions ($n = 8$); 0, 0.15, 0.15, 0.3, 0.3, 0.6, 0.6, and 0 mM over 8 sessions ($n = 16$); or 0, 0.3, 0.3, 0.6, and 0.6 mM over 5 sessions ($n = 4$). In addition, self-administration of AMPA was examined with varying lengths of injection cannulae

along the dorso-ventral axis, to determine boarder between AMPA-effective and -ineffective zones within the septal area. Four to five sites along the vertical axis were examined with the increment of 0.7 mm. At each site, 3 concentrations of AMPA (0, 0.03, and 0.1 mM in this order over 3 sessions) were examined.

*AMPA receptor antagonist.* Twelve rats that had gone through one of the self-administration procedures described above received vehicle, 0.25 mM AMPA alone, and a mixture of 0.1 mM ZK 200775 and 0.25 mM AMPA over 3 consecutive sessions. The order of these treatments was counterbalanced among the rats.

*DA-dependent effects of (S)-AMPA.* Effects of the D1 receptor antagonist SCH 23390 on self-administration of (S)-AMPA into the MS were examined over 3 consecutive sessions. After receiving 0.9% saline (1 mL/kg, i.p.) 30 min before the start of session 1, rats received intracranial vehicle infusions. At 30 min before sessions 2 and 3, the rats were treated with saline and SCH 23390 (0.025 mg/kg, i.p.) and given the opportunity to self-administer 0.25 mM (S)-AMPA. The order of these pretreatments was counterbalanced among the rats.

*Visual-stimulus seeking behavior.* Experimentally naive rats were individually placed in the same operant-conditioning chambers described above except that they were equipped with two levers. Responding on the active lever illuminated the cue light just above the lever for 1 s and turned off the house light for 7 s (during which lever pressing was counted but produced no programmed consequence). Responding on the inactive lever had no programmed consequence throughout the session. The left–right locations of the active and inactive levers were counterbalanced among rats, and the assignment of active and inactive levers remained the same for each rat throughout the experiment. In addition, the number of lever-presses required to produce VS increased by 1 for every 10 VS presentations that the rat earned in order to facilitate differential responding between the two levers. The rats received 0, 0.01, 0.05, 0.1, 0.25, and 0 mM AMPA in this order over 6 sessions. Infusions (75 nL each) were delivered on a fixed 90 s interval schedule. Each session lasted 90 min and sessions were separated by 1 day.

**Mouse ex vivo electrophysiology.** vGlut2-Cre mice received intracranial injections as described above. We injected 150 nL of pAAV1-Ef1a-DIO-hChR2 (H134R)-EYFP-WPRE-pA into the SuM, and 300 nL of AAV1-EF1a-DIO-nuc-tdTom (titer $2.60E + 12$ vg/mL; NIDA OTTC), which expresses Cre-specific nuclear-localized tdTomato fluorophore, into the MS. Mice were deeply anesthetized with isoflurane (60–90 s) and then rapidly decapitated. Coronal slices containing the septum were cut in ice-cold solution containing (in mM) 92 NMDG, 20 HEPES, 25 glucose, 30 NaHCO3, 1.2 NaH2PO4, 2.5 KCl, 5 Na-ascorbate, 3 Na-pyruvate, 2 thiourea, 10 MgSO4, 0.5 CaCl2, saturated with 95% O$_2$, 5% CO$_2$ (pH 7.3–7.4, ~305 mOsm/kg) and incubated for 5–10 min at 35 °C in the same solution. Slices were allowed to recover for a minimum of 30 min at room temperature in aCSF containing (in mM) 126 NaCl, 2.5 KCl, 1.2 MgCl2, 2.4 CaCl2, 1.2 NaH2PO4, 21.4 NaHCO3, 11.1 glucose, 3 Na-pyruvate, 1 Na-ascorbate. Recordings were made at 32–35 °C in the same solution which was bath perfused at 2–3 mL/min. Intracellular solution contained (in mM) 115 K-gluconate, 20 KCl, 1.5 MgCl2, 10 HEPES, 0.025 EGTA, 2 Mg-ATP, 0.2 Na2-GTP, 10 Na2-phosphocreatine (pH 7.2–7.3, ~285 mOsm/kg, Cl-reversal potential ~ −50 mV). Virus-infected cells (Td-tomato+) and axon terminals (eYFP+) were identified using scanning disk confocal microscopy (Olympus FV1000), and differential interference contrast optics were used to patch neurons. Whole cell current-clamp recordings were performed in visually identified neurons in the septum. For ChR2 experiments, a 473 nM laser (OEM laser systems, maximum output 500 mW) attached to a fiber-optic cable was used to deliver light to the slice. Light intensity of 8–12 mW was used to stimulate ChR2-expressing terminals in slice recordings. For experiments measuring optogenetic evokes EPSP, 2 ms light pulses were delivered at a frequency of 0.5 Hz. Following a 10 min baseline period, DNQX (10 μM) and AP5 (50 μM) were added to the recording bath solution. *I–V* curves were not recorded after DNQX/AP5 application. For latency analysis, 5–10 traces from the baseline recording period were averaged, then response latency was defined as the timepoint after the onset of the light stimulus at which the change in membrane potential passed a threshold of 1–5 mV/s, identified using threshold detection, or in some cases manual detection. Recordings were discarded if series resistance or input resistance changed >20% throughout the course of the recording. An Axopatch 200B amplifier (Molecular Devices) and AxoGraph X software (AxoGraph Scientific) were used to record and collect the data, which were filtered at 10 kHz and digitized at 4–20 kHz.

### In vivo electrophysiology
*Surgery.* Mice received injections of Cre-specific ChR2 vector into SuM, were left to recover for 1 week, and then participated in sucrose-seeking training. Mice showing cue discrimination, as determined by greater latency to approach sucrose port for CS − than CS+, then received a second surgery to implant tetrode assemblies into the SuM and an optic fiber into the MS. We used a bundle of four tetrodes (16 channels) for in vivo recording in the SuM. The tetrode bundle was coupled with a movable (screw-driven) microdrive assembly (~1 g weight)[62]. Each tetrode consisted of four wires (90% platinum and 10% iridium; 18 μm diameter with an impedance of ~1–2 MΩ for each wire; California Fine Wire). All mice were then acclimated to the

electrophysiology system tethering, and performed the conditioned operant sucrose seeking for several more sessions until behavioral metrics, e.g., number of rewards earned, latency to approach after CS+ and CS−, had again stabilized.

*Spikes.* Neural signals were pre-amplified, digitized, and recorded using a Neuralynx Digital Lynx acquisition system. Spikes were digitized at 32 kHz and filtered at 600–6000 Hz using one recording electrode that lacked obvious spike signals as the reference. The electrode bundle was lowered by ~80 µm after completing 1 successful recording session. Several depths of recording in the SuM were performed in each mouse, and the recording depth used for data analysis was determined upon histological analysis of tetrode endpoint and the distance the tetrodes were driven. We used multiple spike-sorting parameters (for example, principal component analysis and energy analysis) of Plexon OfflineSorter to isolate recorded spikes. Sorted spikes were processed and analyzed in NeuroExplorer (Nex Technologies) and Matlab (Mathworks). Interspike interval histograms were made to confirm that no discharges occurred during refractory period (<1.0 ms). Low-frequency firing neurons of less than 0.5 Hz were excluded from the study's analyses due to insufficient number of spikes during behaviors of interest.

*Electrophysiological data analyses.* Key analyses on electrophysiological data were based on the sample size of 152 SuM neurons from 6 mice with $25 + 7$ (mean + SEM) neurons per mouse. These numbers are generally considered sufficient in conducting statistical analyses on electrophysiological data. Neurons were categorized into SuM-septum system neurons by whether they were activated upon photostimulation of SuM terminals in the septum. Following each sucrose-seeking behavior session, we delivered 50–75 trains consisting of 15 light pulses (10 ms pulse in duration) at 22 Hz with a variable interval of 10, 12, or 15 s per testing session. To categorize recorded neurons as laser-increase, laser-decrease, or laser-unaffected, we calculated the firing rate of each neuron 2 s directly before each laser pulse train. We used the mean and standard deviation of these values to calculate z-scores for the firing rate of each neuron for 0.5 s after the onset of the laser pulse train. Neurons were classified as SuM:Sept$^{vGlut2}$ circuit-increase if the mean of these z-scores was >1, and SuM:Sept$^{vGlut2}$ circuit-decrease if the mean z-score was < −1. All other neurons were classified as SuM non-laser responsive.

To determine the general event-related activity of all neurons, we generated perievent histograms with 200 ms bins 10 s before and after each event. These data were smoothed using a moving average filter with a span of 3 data points (600 ms) on either side of a given bin. The firing rates for each individual unit were normalized by converting to z-scores calculated using mean baseline firing rates from −10 to −6 s before lever press and nose poke events, and +6 to +10 s after nose poke-withdrawal events. We generated heatmaps using the smoothed and normalized perievent data to visualize the effect of CS+ and CS− cues on SuM neuron firing rates. In the heatmaps, neurons were sorted and displayed by the difference in their firing rates during the 2 s cue period, with neurons having the greatest positive difference between CS+ and CS− (i.e., CS+ higher than CS−) on top and vice versa for neurons responding more to CS− cue on bottom. To easily follow the event-correlated activity for each neuron during sucrose-seeking and cue discrimination, on through sucrose drinking behaviors, we additionally generated heatmaps to visualize sucrose port entries and exits directly after CS+ and CS− events using the same display-order used in the lever-press heatmap. Additionally, neurons were categorized as to whether their mean z-score during CS+ (and again for CS−) increased, decreased, or did not have an absolute value $z > 1$ (i.e., no change). The cumulative traces of their perievent data were plotted on the same time scale as heatmap data.

To examine changes in SuM single-unit activity during grooming behaviors occurring during the sucrose-seeking behavioral assay, we scored video recordings of these sessions for grooming bouts that were longer than 2 s in duration. A trained researcher scored videos based on standard mouse grooming behaviors[63] and perievent heatmaps were created as described above for start and end of grooming bouts. We also analyzed the number of neurons that were sensitive to grooming behavior by calculating firing rate change (either $z > 1$ or $z < 1$) during 2 s before and/or after the start and end of grooming bouts. Not all mice had scorable grooming bouts due to the image-quality issue, so some sessions were excluded from this analysis.

*Locomotor activity and single-unit firing rate correlation.* Correlations of mouse locomotor activity during single-unit recording sessions were determined by offline analysis of overhead video recordings acquired during conditioned operant sucrose-seeking recording sessions. The video and single-unit data were synchronized by the Neuralynx acquisition software. X and Y position of the center of the mouse were determined using ezTrack[64] and all tracks were visually inspected for accuracy. To unbiasedly determine when the mouse was 'near' the side of the chamber that contained the sucrose port and lever, the mean X-position (X-axis being perpendicular to the wall of the chamber that housed the sucrose port and lever) was calculated and used to define epochs where the mouse was either 'near' (X-coordinate < mean X-position) or 'away' (X-coordinate > mean X-position) for time periods ≥2 s. In addition, time periods when the mouse was in the sucrose port, i.e., when the sucrose port IR-beam was broken, were used to determine locomotion during 'near' epochs that excluded consummatory behavior, or 'near, not in sucrose port'. Using X and Y position during 'away', 'near', or 'near, minus in-sucrose port' epochs, distance traveled was calculated, and along with single-unit

spike the data were binned into 2 s time bins and correlation coefficients, R-values, and p-values calculated using Matlab's corrcoef function.

*Histology for tetrode location verification.* At the completion of the electrophysiology recordings, the final electrode position was marked by passing a 20 s 10 µA current using a linear constant current stimulus isolator (Neurolog System) through two tetrodes that showed good spiking data. Mice were deeply anesthetized and intracardially perfused with ice-cold phosphate-buffered saline followed by 10% formalin. Brains were removed and post-fixed in formalin for at least 24 h before being rapidly frozen and sliced on a cryostat (40 µm coronal sections). Sections from the septum and SuM were collected and mounted with the Mowiol mounting medium mixed with DAPI for fluorescent microscopic examination of viral vector expression, as well as optical fiber and tetrode placements.

*Conditioned operant sucrose seeking.* Mice were acclimated to water restriction over several days. Training for sucrose-seeking procedure is as follows.

## First training procedure
Both levers are available throughout the 30 min session. A lever-press results in a 2 s auditory cue (CS+), which begins after the lever-press, followed by delivery of the sucrose solution (~20 µL of 8% sucrose in facility water) at the end of the cue. Additional lever-presses do nothing until mice retrieve the sucrose reward as detected by breaking the photobeams in the sucrose port. Occasionally, sucrose-solution droplets are added to the levers to facilitate approach to the lever and training in general. Mice had to successfully earn ≥21 rewards during the session before moving to the second training procedure.

## Second training procedure
Only one lever (same lever throughout the session) enters the chamber after a random intertrial interval of 25–65 s. Once the lever is presented, the mouse has unlimited time to press the lever which again results in the same CS+ 2 s auditory cue after the lever-press followed by delivery of the sucrose solution at the end of the cue. After the sucrose is delivered, the lever is retracted and presented again after the random intertrial interval. Mice had to successfully earn ≥21 rewards during the session before moving to the conditioned operant sucrose-seeking procedure.

## Conditioned operant sucrose-seeking procedure
This procedure is identical to the second training procedure, except a second auditory cue is added and serves as a predictor of no-sucrose reward (CS−). A no-sucrose-cue is presented after lever-press occur in 50% of the trials, randomly, while the sucrose-reward-cue (CS+) occurs in the other 50% of trials. We record latency to enter the sucrose port after CS+ vs CS− to measure discrimination between the two cues to determine whether the mice learn the predictive nature of the cues. For all mice, either a 5 kHz pure tone or a 5 Hz click served as the CS+ or CS− in a counterbalanced manner.

### Mouse cannula experiments
*Cannula implant.* Surgeries were performed as described above for optogenetic fiber implantation, however, here, a 24-guage guide cannula (Plastics One; Roanoke, VA) was sterotaxically implanted into WT (c57bl/6 J; $n = 11$) mice to a depth of 3.7 mm from skull surface above SuM. We implanted the cannula targeting the SuM in 11 mice. We later removed 5 mice due to the cannula being outside of the SuM. Five out of the 6 mice with cannulas in the SuM successfully met criteria for CS+ vs CS− cue discrimination and received intra-SuM mus+bac or vehicle (sterile saline) in a counterbalanced order. Mice recovered for 1 week before the onset of water restriction and sucrose-seeking training. Dummy cannulas extending just past the end of guide cannula were used and were occasionally removed and replaced during behavioral acquisition training to habituate mice to cannula procedural handling.

*Sucrose seeking and temporary inactivation.* See 'Methods' for 'In vivo electrophysiology' for description of conditioned operant sucrose seeking. Two mice out of 7 failed to learn to lever press for reward, and they were not included in the experiment. Once mice acquired CS+ vs CS− cue discrimination, a baclofen (Tocris Bioscience; #0417) + muscimol (Tocris Bioscience; #0289) cocktail (25 ng baclofen + 25 ng muscimol, dissolved in 0.25 µL sterile saline) was infused via a 31-gauge injection cannula (Plastics One; Roanoke, VA) extending 1 mm past guide cannula tip (into SuM) over 1 min. Cannulas were left in place for an additional minute to allow for diffusion of the drug into brain tissue. Immediately after the infusion, mice were placed into behavioral chambers for the 30 min sucrose-seeking session. Each mouse was tested after receiving the mus+bac cocktail or vehicle (saline) infusions to serve as control. The order of infusions was counterbalanced with a day of a free sucrose-seeking session in between to minimize carryover effects between infusion sessions.

*Sucrose drinking and temporary inactivation.* After mice finished sucrose-seeking testing described above, they were retrained in the operant chamber to receive sucrose solution, ad-lib. The two mice that were excluded from the above experiment were included in this experiment (i.e., $n = 7$). The amount of sucrose consumed, time spent in sucrose port (as measured by photobeams placed just in front of sucrose spigot), and chamber crossing were recorded. Once the daily volume of sucrose consumed

stabilized, mice were again tested using the same counterbalanced mus+bac infusion procedure described above, with a free day of sucrose drinking between mus+bac or saline infusions. Because of video recording issues, one of the animals was not included for the analysis of consummatory responses in 5 min bins.

**Anesthetized mouse fMRI with optogenetic stimulation**

*Animal preparation*. Animal preparation procedures for fMRI in anesthetized rodents were detailed previously[65]. We used a combination of low dose of gaseous anesthesia isoflurane and subcutaneous dexmedetomidine. This protocol has been shown to preserve neurovascular coupling, and the default mode network has been successfully identified with this regime[66]. Anesthesia was induced with 2% isoflurane inside an induction chamber, and dexmedetomidine was administered at 0.01 mg/kg (i.p.) (Zoetis, Parsippany, NJ). Mice were subsequently placed onto a customized holder in the prone position, with their heads secured in place via a nose-cone bite bar and ear bars. During scans, rodent's rectal temperature, respiratory rate and non-invasive pulse oximetry and heart rate (SA Instruments, Inc., NY, USA) were monitored. Body temperature was maintained at $37.1 \pm 0.5\,°C$ using a water-circulating heating pump. Prior to functional scans, isoflurane was systematically reduced to about 0.5%, while dexmedetomidine (0.015 mg/kg/h) was continuously infused subcutaneously. The level of hemoglobin oxygen saturation ($SpO_2$) was maintained above 96% by adjusting the oxygen concentration in the inhaled gas mixture (oxygen + air).

*MRI scan*. MRI data were acquired with a 9.4T Bruker Biospin scanner equipped with an active-shielded gradient coil, a quadrature volume coil for radiofrequency (RF) excitation, and a single-turn 2 cm diameter surface coil for MR signal reception (Bruker Medizintechnik, Karlsruhe, Germany). High-resolution T2-weighted anatomical images were acquired with the following scan parameters: TR = 2200 ms, TE = 35 ms, field of view (FOV) = $30 \times 30$ mm$^2$, matrix size = $256 \times 256$, with 25 slices acquired at a slice thickness of 0.6 mm. Functional images were acquired using a single-shot gradient echo-planar imaging (EPI) sequence with the following scan parameters: TR = 1000 ms, TE = 15 ms, FOV = $25 \times 15$ mm$^2$, matrix size = $96 \times 58$, with 15 slices acquired at a slice thickness of 0.6 mm. Distortions in EPI images were corrected using a reversal k-space strategy[67].

*fMRI data analysis*. The imaging experiment consisted of a block design paradigm with a 20 s baseline followed by five cycles of 20 s ON and 40 s OFF, each ON epoch consisted of continuous trains of photostimulation (one pulse train per second, each train consisted of 15 pulses delivered at 25 Hz with a pulse duration of 5 ms). Laser intensity was set to deliver 10 mW of 473 nm light at the tip of the optic fiber in the brain. fMRI data were analyzed using the ANFI software package (V17.0.16)[68]. Image processing steps are as follows: High-resolution anatomical images were co-registered onto a template using AFNI function *3dAllineate*. fMRI time series were then co-registered using the *3dvolreg* function. EPI voxels were resampled to a larger voxel size (0.375 mm × 0.375 mm × 0.6 mm) and spatially normalized using a full-width at half-maximum kernel of 0.5 mm to increase the signal-to-noise ratio (SNR). BOLD activation was quantified as follows: The experimental block design was convolved with the canonical hemodynamic function using AFNI's waver function to model the hypothesized ideal BOLD response; and the motion-corrected preprocessed fMRI data were deconvolved with the ideal BOLD time course with the six motion parameters (x-, y-, z-translation, yaw-, pitch-, roll-rotation) used as nuance variables. Output data from *3dDeconvolve* were fed into a generalized least squares time series fit using AFNI's *3dREMLfit* to calculate the correlation, beta-weights, and associated t-value in relation to the ideal BOLD response. Group activation maps were derived through a voxel-wise beta-weighted t-test against zero using *3dttest*. AFNI's function 3dcluststim was used to control for false discovery rate and estimate statistical significance level with a cluster size of 20 voxels. This resulted in a $P_{corrected} < 0.01$ for a voxel to be considered significant. Activation maps were overlaid onto high-resolution anatomical images for display.

**Statistical analysis**. Data were analyzed using Statistica (version 6.1, StatSoft, Inc., Tulsa, OK, USA), Graphpad Prism (GraphPad Software, La Jolla, CA, USA) and MATLAB (The MathWorks, Inc., Natick, MA, USA). Specific procedures for statistical tests are described thought the 'Results' section and figure legends.

**Reporting summary**. Further information on research design is available in the Nature Research Reporting Summary linked to this article.

## Data availability
The data that support the findings of this study are available from the authors on reasonable request, see author contributions for specific data sets. The MATLAB code that we used are reposited in Github: https://github.com/sikemoto/photometry-recording-w-stim/tree/main; see the code named 'OptoStimRecording.m'. Source data are provided with this paper.

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

## Acknowledgements

The present work was supported by the Intramural Research Program of National Institute on Drug Abuse, National Institutes of Health. We thank the NIDA-IRP Optogenetics and Transgenic Technology Core and the Stanford Optogenetic Innovations lab for providing viral vectors and vector plasmids, respectively. We also thank Deon Harvey for editorial assistance.

## Author contributions

Conceptualization, A.J.K. and S.I.; Methodology, A.J.K. and S.I.; Investigation – Behavior, A.J.K., R.S., C.B.C., C.T.P., R.F.D., S.J., A.F.A., C.G.C. and L.A.R.; Investigation – In vivo electrophysiology, A.J.K.; Investigation – Ex vivo electrophysiology, L.A.R.; Investigation – fMRI, C.G.C.; Writing – Original Draft, A.J.K. and S.I.; Writing – Review & Editing, A.J.K., R.S., C.B.C., C.T.P., R.F.D., S.J., A.F.A, C.G.C., D.V.W., L.A.R., H.L., Y.Y. and S.I.; Funding, L.A.R., Y.Y. and S.I.; Supervision, D.V.W., H.L. and S.I.

## Funding

## Competing interests

The authors declare no competing interests.
