## [Peer Review File · Nature Communications]

Reviewers' comments:

Reviewer #1 (Remarks to the Author):

The Authors demonstrate that activation of the supramammillary region (SR), in which almost or all neurons are glutamatergic, facilitates anticipatory behavior. They suggest that this is mediated by a pathway that involves the medial septum and involves both glutamate and dopamine. This review focuses on major problems.

Overall the paper includes important data and experiments, but there are some conceptual, evidential and methodological problems. As it stands, I feel that the paper may not add a clear contribution that can be reconciled with other extant literature. The principal concern is that no mechanism and pathway is described for SR downstream targets to interact with the canonical reward networks.

Major Concerns

1. Conceptual

1a. The abstract commences with the statement that the SR 'regulates reward and anticipatory behavior'. This is based in the corresponding author's prior work. The area called the SR in these papers includes a rostral and medial portion that is outside of the non-GABAergic portion of the SR and more likely related to the posterior hypothalamic nucleus (PH). The supramammillary GABAergic population elevates along the mammillothalamic tract but principally projects to the hippocampus (see below) and the cannula sites in these papers are within and medial to this population, as well as in the SR proper. It is not clear to what extent significance is driven the rostral-medial cell population that is not part of the non-GABAergic SR versus the SR. Consistent with this critique is the finding that activation of the PH, of which the above neurons are likely a member, is also rewarding (Ikemoto 2004 AMPA paper). While these results may be consistent with SR activation being rewarding, it does not indicate that the SR is a critical component of the 'reward system', particularly in the SR proper glutamate neurons.

1b. While perhaps somewhat pretentious and beyond the scope of the paper and this review, it may be helpful to adopt an eliminativist perspective and keep in mind that whole animal experiments with psychological constructs like reward and anticipation may not map neatly on to specific neural circuits and a neat circuit is not fully elucidated here (see below). The statement that the SR regulates the septohippocampal system is certainly better established and a role in the regulation of REM sleep and arousal also seems likely and is consistent with some results (see below).

1c. The literature cited about theta activity is mostly from Vanderwolf, a true giant of this field, but does not take into account more modern perspectives. Causality is implausibly implied in the sentence on page 9, top, that 'hippocampal theta oscillations, which regulate voluntary environmental interaction, thereby anticipatory behavior' Theta oscillations do not cause or regulate anything per se, nor is the correlated state of the septohippocampal circuit required for voluntary action. This seems a very questionable explanation of the author's findings in terms of the literature pertaining to SR regulation of the septohippocampal system. Similarly, explanations about reward and anticipatory behavior being due to the septohippocampal projections seem problematic.

1d. Is it possible that the facilitation of anticipatory behavior was mediated by an increased level of

attention or arousal? It would be helpful to discuss this point in the context of other recent papers about the SR. The decrease locomotion seen with SR inhibition (experiment 8) is consistent with drowsiness, but given no anatomical substrate, does not seem consistent with a specific motor effect nor one mediated by reward (see below regarding anatomical projections).

1e. There is no 'dopamine system' per se, but multiple groups of dopamine neurons that have distinct roles that are much broader than reward.

1f. Overall, the mechanism is unclear - please see evidential points.

1g. Discussion of reward, anticipation and consummation in the paper are interesting.

2. Evidential

2a. Medial septal glutamate neurons do not project into the reward system. Instead they principally project to brainstem nuclei (particularly the dorsal raphe and nucleus incertus), the SR with light projections into the hippocampal formation (Agostinelli et al., 2019). While some neurons project to the horizontal diagonal band, these also do not innervate the reward system. How is it that the MS activates the nucleus accumbens? This is not clarified at all, but, based on the work presented, suggests that neurons outside of the SR and MS were included in manipulations/injection sites (see methodological below regarding microinjection technique). The pattern of LH and midline thalamus BOLD activation on fMRI makes some sense in terms of known MS glutamate neuron projections (see Agostinelli also). The interaction of SR  MS and VTA  NAcc circuits remains unclear and needs to be clarified for the work to be acceptable in my view.

2b. There are more neuronal populations in the SR and DA neurons are caudal and few. NOS neurons have also been identified (Wisden group, Fuller group). NOS neurons may have a role in REM, as does the SR in general, so dissecting neuronal populations seems critical to understanding this brain region, as does a consideration of topography within the SR.

3. Methodological

3a. It is impossible to make small injections that obey the fields of discussed nuclei with the technique described (pressure injection through a ~185 um needle. Thus, all injection sites, infusion sites (as in Figure 3) and estimated areas of optogenetic activation (as in S1, S2, S3) for all mice/rats and experiments should be plotted in supplementary material. The misplacement or spread of injections, infusions could account for co-activation of reward pathways which do not seem to be connected to SR or MS projections.

3b. Noting the above, some red optogenetic areas of activation in supplementary figures appear to be in the PH and the rostral-medial portion of the SR which is a questionable part of the nucleus.

Reviewer #2 (Remarks to the Author):

The study by Kesner and colleagues examines the role of circuits centered on the supramammillary nucleus (SuM) in reward-guided behavior. In a long series of experiments, the authors show that SuM glutamatergic neurons projecting to septum glutamatergic neurons can support intracranial self-

stimulation. Cells in the SuM had diverse responses during a rewarded v unrewarded cue presentation, but uniformly inhibited during reward delivery. Inactivation of SuM disrupted all locomotor activity in an operant task, though didn't change the total amount of sucrose consumed in a sucrose drinking situation (only the bout length). Systemic administration of dopamine antagonists disrupted ICSS, which the authors take as evidence of interactions between SuM and dopamine transmission. Finally, the effects of optogenetic stimulation of SuM-to-septum or dopamine pathways from VTA to ventral striatum on whole brain BOLD responses was contrasted.

Within this manuscript is a potentially interesting dissection of a circuit that was new to me and how it may be involved in reward-guided behavior. However, the sheer weight of experiments described here makes it a real slog to get through. This is in part as sometimes the reasoning behind particular manipulations is hard to work out (e.g., lever pressing for lights), in part as the experiments themselves sometimes muddy the narrative that the authors are trying to bring to the paper. For instance, the electrophysiological responses of SuM neurons do not look like a population important for regulating anticipatory responses (Experiment 7); numerically more increase their activity after the CS- than a CS+. In fact, there are few consistent increases in activity at all, which somewhat calls into question what can be taken from a series of experiments using optogenetic activation.

Therefore, although I find the work of potential interest, I am concerned in its present form that the manuscript cannot say much more than this circuit is somehow involved in mediating reward-guided behavior. My specific comments are below:

1. The authors have obviously tried to be specific about what they mean by "reward that triggers anticipatory behavior". Nonetheless, in the context of these experiments, it remained unclear what precisely this function is. For instance, the majority of the experiments focus on optogenetic activation of specific SuM neurons or projections, yet, as mentioned above, it doesn't seem to differentially encode responses after a CS+ or CS- as would be expected for a structure to "mediate anticipatory behavior" (p3 Intro). Indeed, from these plots, it seems as if the primary factor SuM is signaling is reward delivery and that is signaled by an inhibition of activity. Moreover, in the inactivation experiments, all locomotor activity, not just anticipatory reward-seeking movements, is reduced.

Therefore, the authors first need to be more precise in describing what they can and cannot conclude from the data. E.g., given that during sucrose delivery, the mice will be remaining in one place, can they show that SuM neurons do not just care whether the animal is moving or not? In other experiments, can the authors rule out more basic changes in locomotor activity for their effects? Can the authors say whether their proposed function is an all-or-nothing effect or is instead graded based on the prediction of future reward? Does it predict the vigor of the response on a trial-by-trial basis?

Moreover, they need to consider explicitly the implications of probing a circuit using optogenetic activation when the key activity patterns are inhibitions of activity.

2. I was confused by the ICSS task having an additional light presentation for active responses,

particularly given the later experiment which explicitly used light presentation as a reinforcer. Surely this just adds an extra factor that could be reinforced? I.e., it could be the action or the cue that is being reinforced or a combination that is being reinforced.

3. On a related note, please explain the logic for why they test visual seeking behavior? Are the authors arguing that a light presentation substitutes for reward here? If not, what does it say about the hypothesis of SuM mediating reward that triggers anticipatory behavior?

4. Many of the groups increase pressing with optogenetic stimulation (or sustain pressing above baseline), even if not to the same extent as the SuM:Sub glutamate groups (e.g., Fig 1D, Fig 2A). What is the interpretation of this?

5. The EPSPs in TD-tomato positive neurons in Figure 4g looked much more equivocal than the text suggested. The “representative” plot in 4f does not look very representative of the bar plot, where there only appears to be 1 highly activated cell and several around zero.

6. It wasn't clear to me at what point during the operant sucrose seeking task the recordings were taken. This is important given the animals' behavior was changing across the sessions. Depending on the session, the data time-locked to entering the sucrose port after the CS- might mean very different things.

Minor comments

- The labeling of figure 2 does not match the main text.
- Please somewhere note numbers of neurons recorded / animal.
- Why were the dopamine antagonist doses not counterbalanced across animals?
- P32 – say how much the laser power was increased / decreased for the mice in the SuM:Sept and VTA:VStr mice
- There are many more refined protocols for rat anesthesia than the one used here, particularly given the potential for irritation of chloral hydrate administered ip. I would advise the group to update their anesthetic regime for rats if they haven't already done this. Also, can the authors please confirm that their local IACUC approved this regime.
- P5, line 155, F or p value must be wrong
- Is the fMRI experiment really just a “pilot” experiment (p11, line 327)?

Reviewer #3 (Remarks to the Author):

In this study Kesner et al. are reporting a careful analysis of the role of the supramammillary nucleus (SuM) and glutamatergic projections of this nucleus on operant reward conditioning of mice. Although highly interesting and relevant for the field of circuit neuroscience, the organization and presentation of

the results appear not as rigorous as the realization of the experiments. Moreover the ex vivo physiology experiment do not support the conclusions of a monosynaptic connection as claimed by the authors.

First, the title, abstract and results state a role for SuM neurons in anticipatory behaviors. However, the experiments are very specific of reward anticipation, and the selectivity of this study for positive valence should be clarified starting in the title of the manuscript.

Second, the Figure 6 describing general inhibition of the SuM (no projection specificity) would allow a better narrative if it was the second Figure, before analysis of the influence of different projections of the SuM.

Third, in the ex vivo electrophysiological experiments, the authors claim they identified a monosynaptic connection. However, they did not block recurrent excitation and the current they observe in baseline condition (ACSF) might be due to polysynaptic excitation. To unequivocally demonstrate a monosynaptic connection, the authors should use ChR2-assisted circuit mapping (CRACM) by applying TTX and 4AP on the sample.

Finally, from the in vivo electrophysiology recordings (tetrodes, Figure 5), the author conclude that the SuM neurons are inactive during reward consumption. However, the data represented are changes in firing rate, and a decrease of firing does not necessarily reflect that the neurons are inactive. The authors should either rephrase their conclusions, or represent the absolute firing rates in the figure, rather than z-scores.

----- Minor Comments -----

The authors seem to interchange reward and reward seeking, which is inaccurate and confusing. In the abstract the author mention "SuM-Septum reward", when what they mean is likely reward seeking, or another reward-related behavior. The reward is the reinforcer (sucrose consumption, or optogenetic manipulation of a neuron population) not the behavioral response or state. Similarly in the introduction (line 44) the authors say "Such brain manipulations must elicit a transient state, referred to as reward, that triggers goal-directed information processing for foraging or other seeking behavior; that is, anticipatory behavior." This definition attempt is very unclear. Do the authors mean 'reward state' rather than reward (which is the reinforcer) ? or maybe they mean reward expectation ? reward anticipation ? It is crucial to correct and clarify these definitions.

The brain schematic representing the location of viral injection and optic fiber placements are hardly readable as they are very small (Figure 1A, 2A, 2E...). An expanded schematic (or just with 3 boxes: SuM, septum, vSub) would facilitate understanding of the experimental design.

Numbering the experiments in the titles of the sections is not informative and distracting for the reader. Please remove this part of the title.

Figure 1 H: the author mention 50 nL in the Figure, but the meaning (volume of viral vector injected) is

not explained in the legend. Please specify this in the legend.

The font of the figures is sometimes Arial and sometime Calibri. Aesthetic would be better with a uniform font.

Reviewers' comments:

Reviewer #1 (Remarks to the Author):

The Authors demonstrate that activation of the supramammillary region (SR), in which almost or all neurons are glutamatergic, facilitates anticipatory behavior. They suggest that this is mediated by a pathway that involves the medial septum and involves both glutamate and dopamine. This review focuses on major problems.

Overall the paper includes important data and experiments, but there are some conceptual, evidential and methodological problems. As it stands, I feel that the paper may not add a clear contribution that can be reconciled with other extant literature. The principal concern is that no mechanism and pathway is described for SR downstream targets to interact with the canonical reward networks.

The authors' response:

We have rewritten the entire introduction to better provide background information. Particularly, we worked on conceptual issues, to make the present findings easier to understand. The initial key findings of the present study are the SuM-to-MS glutamatergic pathway for motivation and reinforcement. We have conducted additional experiments, which now provide clear evidence that MS-to-VTA glutamatergic pathway relays the motivational effects of the SuM-to-MS glutamatergic pathway to the canonical reward system, i.e. the VTA-VStr dopamine pathway (Fig. 9).

Major Concerns

1. Conceptual

1a. The abstract commences with the statement that the SR 'regulates reward and anticipatory behavior'. This is based in the corresponding author's prior work. The area called the SR in these papers includes a rostral and medial portion that is outside of the non-GABAergic portion of the SR and more likely related to the posterior hypothalamic nucleus (PH). The supramammillary GABAergic population elevates along the mammillothalamic tract but principally projects to the hippocampus (see below) and the cannula sites in these papers are within and medial to this population, as well as in the SR proper. It is not clear to what extent significance is driven the rostral-medial cell population that is not part of the non-GABAergic SR versus the SR. Consistent with this critique is the finding that activation of the PH, of which the above neurons are likely a member, is also rewarding (Ikemoto 2004 AMPA paper). While these results may be consistent with SR activation being rewarding, it does not indicate that the SR is a critical component of the 'reward system', particularly in the SR proper glutamate neurons.

The authors' response:

The reviewer is correct to mention that the lateral portion of the SuM where GABAergic neurons are located largely project to the hippocampus. Our colleague Morales's group (Root et al. 2018, Cell Report) found that about 35% of SuM neurons contains both glutamate and GABA, and they are localized in the lateral to the principal mammillary tract, while the medial SuM contains either glutamatergic or GABAergic neurons with little co-localized neurons. We now provide a new data showing that SuM neurons projecting to the MS are glutamatergic and that MS-projecting glutamatergic neurons are densely distributed in the SuM and scarcely distributed in the posterior hypothalamic nucleus (Fig. 4d-f).

1b. While perhaps somewhat pretentious and beyond the scope of the paper and this review, it may be helpful to adopt an eliminativist perspective and keep in mind that whole animal experiments with psychological constructs like reward and anticipation may not map neatly on to specific neural circuits and a neat circuit is not fully elucidated here (see below). The statement that the SR regulates the septohippocampal system is certainly better established and a role in the regulation of REM sleep and arousal also seems likely and is consistent with some results (see below).

The authors' response:

We agree with the reviewer that the notion of arousal may describe the functional role of SuM neurons. However, the arousal concept is not sufficient for our observations suggesting SuM's roles in motivation and reinforcement. One possibility is that there are many kinds of arousal

and that the SuM-to-MS pathway is involved in the arousal that energizes the host for interacting with the environment in an active, approach manner as opposed to a passive, avoidance manner. We now discuss this idea in a clearer manner. See pages 11-12.

1c. The literature cited about theta activity is mostly from Vanderwolf, a true giant of this field, but does not take into account more modern perspectives. Causality is implausibly implied in the sentence on page 9, top, that 'hippocampal theta oscillations, which regulate voluntary environmental interaction, thereby anticipatory behavior' Theta oscillations do not cause or regulate anything per se, nor is the correlated state of the septohippocampal circuit required for voluntary action. This seems a very questionable explanation of the author's findings in terms of the literature pertaining to SR regulation of the septohippocampal system. Similarly, explanations about reward and anticipatory behavior being due to the septohippocampal projections seem problematic.

The authors' response:

Thanks for pointing out mischaracterizing the role of hippocampal oscillations in behavior. We meant that theta oscillations reflect cognitive processing occurring during active environmental interaction. We have rewritten the sentence (see the 1st paragraph on page 8).

1d. Is it possible that the facilitation of anticipatory behavior was mediated by an increased level of attention or arousal? It would be helpful to discuss this point in the context of other recent papers about the SR. The decrease locomotion seen with SR inhibition (experiment 8) is consistent with drowsiness, but given no anatomical substrate, does not seem consist with a specific motor effect nor one mediated by reward (see below regarding anatomical projections).

The authors' response:

Thanks for pointing out conceptual difficulty. We did not mean that the SuM is involved in anticipatory process per se, but an arousal that invigorates anticipatory or approach behavior. We tried to convey the notion with the phrase "a transient state, referred to as *reward*, that triggers goal-directed information processing for foraging or other seeking behavior; that is, anticipatory behavior." But this description appears not to be easily conceived, partly because of the term "anticipatory", which is a high order process beyond the SuM's role and partly because of the term "reward", which can mean different notions depending on the reader. We have clarified these issues in our revised introduction section.

Although decreased locomotion after intra-SuM injections of muscimol/baclofen is consistent with the notion of drowsiness, the following observations are not: The mice went to sucrose port faster and drank more sucrose when they received muscimol than saline. These observations are best explained by the idea that there are many types of arousal and that the arousal of SuM enables the host to interact with the environment while keeping the host capable of consummatory behavior. The observation supports the idea that approach behavior is regulated by a motivational system distinct from those of consummatory responses. We have revised the discussion section, to make our point more explicit.

1e. There is no 'dopamine system' per se, but multiple groups of dopamine neurons that have distinct roles that are much broader than reward.

The authors' response:

Our initial manuscript defined *reward* as induced state that triggers goal-directed information processing for foraging or other seeking behavior seems to be confusing to many readers, and we therefore rewrote the manuscript without using this term in this sense. Here is our thought on the role of dopamine in reinforcement. Indeed, dopamine neurons are not functionally homogeneous as exemplified by the recent study by Englehard et al. (2019 Nature), which showed various clusters of dopamine neurons respond differently to sensory, motor, and cognitive variables during an approach task. However, these dopamine neurons of different clusters do respond to reward, suggesting a common functional role. Therefore, while some functions are mediated by subsets of dopamine neurons, while other functions are shared by majority of dopamine neurons. Moreover, there is additional evidence to suggest that reinforcement is a function shared among many dopamine neurons. While differential functions between the VTA and substantia nigra (SN) have been reported over the decades, our group found that optogenetic stimulation of SN dopamine neurons reinforce responding as strongly as that of VTA dopamine neurons (Illango et al. 2014, J Neurosci). Moreover, Saunders et al. (2018, Nat Neurosci)'s findings are consistent with the notion suggested by the Englehard et al. study. They found that rats can learn to display conditioned approach to the cue that has been paired with optogenetic stimulation of dopamine neurons projecting to the nucleus accumbens (NAc) core, but not NAc shell or dorsal striatum. However, the stimulation is reinforcing regardless of projection sites: NAc core, NAc shell or dorsal striatum. Thus, self-stimulation function appears to be a common property of distinct functional populations of dopamine neurons. Given these findings in dopamine neurons, it is reasonable to speculate that SuM neurons most likely play distinct roles from dopamine neurons in the aspect of approach behavior; yet SuM neurons share common functions of reinforcement and motivation with dopamine neurons.

1f. Overall, the mechanism is unclear - please see evidential points.

The authors' response:

We now provide the "mechanism" with fiber photometry experiments. Figs. 8, S8 and S9 show the findings that the VTA-to-ventral striatum dopamine neurons are activated by optogenetic stimulation of SuM-to-MS glutamatergic neurons, MS glutamatergic neurons and MS-to-VTA glutamatergic neurons. Thus, these results provide strong evidence that the SuM-to-MS glutamatergic pathway activates the VTA-to-ventral striatum dopamine pathway via the MS-VTA glutamatergic pathway.

1g. Discussion of reward, anticipation and consummation in the paper are interesting.

The authors' response:

Thanks! We believe that our revised discussion is more interesting and informative.

2. Evidential

2a. Medial septal glutamate neurons do not project into the reward system. Instead they principally project to brainstem nuclei (particularly the dorsal raphe and nucleus incertus), the

SR with light projections into the hippocampal formation (Agostinelli et al., 2019). While some neurons project to the horizontal diagonal band, these also do not innervate the reward system. How is it that the MS activates the nucleus accumbens? This is not clarified at all, but, based on the work presented, suggests that neurons outside of the SR and MS were included in manipulations/injection sites (see methodological below regarding microinjection technique). The pattern of LH and midline thalamus BOLD activation on fMRI makes some sense in terms of known MS glutamate neuron projections (see Agostinelli also). The interaction of SR  MS and VTA  NAcc circuits remains unclear and needs to be clarified for the work to be acceptable in my view.

The authors' response:

We appreciate the information. We have added the results of our tracing experiment, which shows that MS glutamatergic neurons project to the paranigral zone of the VTA with unequivocal presence of terminal boutons (Fig. 8c). This results with those of our fiber photometry experiments now clearly show the pathways through which SuM glutamatergic neurons can modulate the activity of dopamine neurons.

2b. There are more neuronal populations in the SR and DA neurons are caudal and few. NOS neurons have also been identified (Wisden group, Fuller group). NOS neurons may have a role in REM, as does the SR in general, so dissecting neuronal populations seems critical to understanding this brain region, as does a consideration of topography within the SR.

The authors' response:

We appreciate the reviewer's suggestion. However, we feel that the role of NOS neurons in reward is beyond the scope of our study.

3. Methodological

3a. It is impossible to make small injections that obey the fields of discussed nuclei with the technique described (pressure injection through a ~185 um needle. Thus, all injection sites, infusion sites (as in Figure 3) and estimated areas of optogenetic activation (as in S1, S2, S3) for all mice/rats and experiments should be plotted in supplementary material. The misplacement or spread of injections, infusions could account for co-activation of reward pathways which do not seem to be connected to SR or MS projections.

The authors' response:

We appreciate the reviewer's concerns about anatomical specificity of micro-infusions and optogenetic stimulation. Because this region is small and difficult to target in general, in preparation of our original manuscript we included several "site control" experiments to address this concern. In experiment 1 (Fig1a-e, S1), we acknowledged that viral transduction and light can spread into adjacent tissue. We therefore maintained the same injection site, but varied optic fiber placements both ventrally and laterally to the SuM – as spread of viral vector and, in particular, light from the optic fiber tip mainly occurs in these directions. We feel the site-level press rate map in S1 indicates that indeed, stimulation of SuM neurons is reinforcing, and that this reinforcement is not due to the spread of stimulation to adjacent areas such as VTA and mammillary body. Indeed, our new experiment with a retrograde tracer shows that glutamatergic

neurons that project to the MS is selectively expressed in the SuM with minor projections from the posterior hypothalamic region.

In experiment 4, we utilized the larger brain of the rat and tested over 100 rats to survey the septal region by systematically varying the canula infusion site and drug concentration. We feel the systematic nature of this study appropriately illustrates that when the injection site is located in the medial part of the septal area (i.e. medial septal nucleus) that the rats engage in ICSS to stimulate neurons in this region with AMPA. We are not claiming that other sites like lateral septum do not support ICSS of AMPA, but that stimulation of medial septal neurons by AMPA action is reinforcing and supports operant responding. This tested and confirmed the hypothesis that SuM glutamate neurons stimulate MS neurons via AMPA-receptor-mediated glutamate transmission. In sum, we believe that it is crucial to show the sites of stimulation as shown in Fig. 3a. In addition, MS glutamatergic neurons are only found along the midline of the septal area. Therefore, we are confident that SuM glutamatergic neurons projecting to the MS mediate positive reinforcement.

3b. Noting the above, some red optogenetic areas of activation in supplementary figures appear to be in the PH and the rostral-medial portion of the SR which is a questionable part of the nucleus.

The authors' response:

In the site-lever press maps in supplemental figures we plotted the tip of the optic fiber (rectangle) with the approximate area affected by illumination. It is an optogenetic technique to place the tip of the optic fiber just above the area of interest (as described in our methods), so the rectangles are indeed mostly in the border between SuM and ventral-medial PH. We do not claim that stimulation of PH is not reinforcing, rather that the majority of our optic fiber placements were placed to stimulate primarily SuM neurons.

Reviewer #2 (Remarks to the Author):

The study by Kesner and colleagues examines the role of circuits centered on the supramammillary nucleus (SuM) in reward-guided behavior. In a long series of experiments, the authors show that SuM glutamatergic neurons projecting to septum glutamatergic neurons can support intracranial self-stimulation. Cells in the SuM had diverse responses during a rewarded v unrewarded cue presentation, but uniformly inhibited during reward delivery. Inactivation of SuM disrupted all locomotor activity in an operant task, though didn't change the total amount of sucrose consumed in a sucrose drinking situation (only the bout length). Systemic administration of dopamine antagonists disrupted ICSS, which the authors take as evidence of interactions between SuM and dopamine transmission. Finally, the effects of optogenetic stimulation of SuM-to-septum or dopamine pathways from VTA to ventral striatum on whole brain BOLD responses was contrasted.

Within this manuscript is a potentially interesting dissection of a circuit that was new to me and how it may be involved in reward-guided behavior. However, the sheer weight of experiments described here makes it a real slog to get through. This is in part as sometimes the reasoning

behind particular manipulations is hard to work out (e.g., lever pressing for lights), in part as the experiments themselves sometimes muddy the narrative that the authors are trying to bring to the paper. For instance, the electrophysiological responses of SuM neurons do not look like a population important for regulating anticipatory responses (Experiment 7); numerically more increase their activity after the CS- than a CS+. In fact, there are few consistent increases in activity at all, which somewhat calls into question what can be taken from a series of experiments using optogenetic activation.

Therefore, although I find the work of potential interest, I am concerned in its present form that the manuscript cannot say much more than this circuit is somehow involved in mediating reward-guided behavior. My specific comments are below:

The authors' response:

We appreciate your comments on narrative. We have rewritten the entire manuscript and made our best effort to make it easier to read through. We realized that the term “anticipatory behavior” was confusing. We did not mean that the SuM is involved in anticipatory process per se, but approach behavior, which can lead to rewards. To avoid confusion, we rewrote the manuscript using new terminology.

1. The authors have obviously tried to be specific about what they mean by “reward that triggers anticipatory behavior”. Nonetheless, in the context of these experiments, it remained unclear what precisely this function is. For instance, the majority of the experiments focus on optogenetic activation of specific SuM neurons or projections, yet, as mentioned above, it doesn't seem to differentially encode responses after a CS+ or CS- as would be expected for a structure to “mediate anticipatory behavior” (p3 Intro). Indeed, from these plots, it seems as if the primary factor SuM is signaling is reward delivery and that is signaled by an inhibition of activity. Moreover, in the inactivation experiments, all locomotor activity, not just anticipatory reward-seeking movements, is reduced.

Therefore, the authors first need to be more precise in describing what they can and cannot conclude from the data. E.g., given that during sucrose delivery, the mice will be remaining in one place, can they show that SuM neurons do not just care whether the animal is moving or not? In other experiments, can the authors rule out more basic changes in locomotor activity for their effects? Can the authors say whether their proposed function is an all-or-nothing effect or is instead graded based on the prediction of future reward? Does it predict the vigor of the response on a trial-by-trial basis?

The authors' response:

We have revised the introduction section on conceptual issues and clarified what we meant by “reward that triggers anticipatory behavior”. Particularly, we replaced the term “anticipatory behavior” with “approach behavior” to eliminate confusion. We do not think that SuM neurons are involved in prediction process per se, but they most likely energize approach-related processes such as action, attention and learning. Particularly, we think that SuM neurons are active when animals need to be physically and mentally attentive to the environment for adaptive approach responses. We now provide additional background information in the introduction section, to help readers to understand the experimental rationale and data interpretations. In

addition, we have provided new analysis on the in vivo electrophysiology data, showing that locomotor activity is not correlated to a great deal with unit activity (Fig S6).

Moreover, they need to consider explicitly the implications of probing a circuit using optogenetic activation when the key activity patterns are inhibitions of activity.

The authors' response:

We believe that there is important merit to use activation to probe functions of the SuM. The abused substance nicotine can directly activate SuM neurons and produces reinforcing effects (Ikemoto 2006 J Neurosci). In addition, the stimulation of VTA DA neurons possibly activates SuM neurons, and this implies that other abused substances that activates VTA DA neurons may activate SuM neurons. Therefore, it is important to understand how the activation of SuM neurons alters the brain and behavior.

2. I was confused by the ICSS task having an additional light presentation for active responses, particularly given the later experiment which explicitly used light presentation as a reinforcer. Surely this just adds an extra factor that could be reinforced? I.e., it could be the action or the cue that is being reinforced or a combination that is being reinforced.

The authors' response:

Our operant conditioning procedure is influenced by the intravenous drug self-administration procedure, in which a cue is typically presented to facilitate learning of the relationship between lever pressing and consequences. While we provided a cue during the acquisition and extinction phases, we did not provide the cue for the reinstatement phase, to confirm that the cue is not required for mice to respond for brain stimulation. We now describe the rationale in the method section (see page 28).

3. On a related note, please explain the logic for why they test visual seeking behavior? Are the authors arguing that a light presentation substitutes for reward here? If not, what does it say about the hypothesis of SuM mediating reward that triggers anticipatory behavior?

The authors' response:

The presentation of unconditioned, visual-stimulus (VS) is known to serve as a reward. We took advantage of this properties of VS to investigate whether increased AMPA transmission in the septum can facilitate on-going reward-seeking behavior reinforced by VS. We used VS because the canonical reward food elicits consummatory responses, which suppress approach-behavior. We asked this question based on the literature on dopamine functions, suggesting that increased dopamine transmission increases approach behavior triggered by cues associated with food, abused drugs as well as unconditioned VS (e.g. Shin et al. 2010, PLOS ONE 5, e8741; Keller et al. 2014, Psychopharmacology 231, 825-840). We re-wrote the introduction section and the result section, to provide clearer background information (2nd paragraph on page 3; 2nd paragraph on page 6).

4. Many of the groups increase pressing with optogenetic stimulation (or sustain pressing above baseline), even if not to the same extent as the SuM:Sub glutamate groups (e.g., Fig 1D, Fig 2A).

What is the interpretation of this?

The authors' response:

For Fig. 1d, we injected the AAV into the SuM and implanted an optic fiber in the SuM, the mammillary body (MB) or the VTA. This is to determine whether light delivered at the SuM affected the MB or VTA. The observation that light delivered at the MB or VTA increased lever pressing slightly over the baseline can be explained either that the light primarily activate the MB or VTA for reinforcement or that the light weakly activated SuM neurons, in which ChR2 was primarily expressed, for reinforcement. These results are not useful in determining whether the stimulation of MB or VTA is reinforcing because the experiment is not designed to examine this question. However, the results of this experiment support the idea that the stimulation of the SuM is reinforcing.

As for the SuM:vSub glutamatergic group (Fig. 2A), when the mice were trained to self-administer vSuB stimulation, they did not increase lever pressing over baseline. An elevated lever pressing with vSuB stimulation was observed after the mice were first trained to self-administer septal stimulation. Therefore, it is most likely that elevated lever pressing with vSuB stimulation was due to a carry-over effect from having the mice previously self-stimulated for septum pathway. Having this said, we initially hypothesized that the stimulation of the SuM:vSuB pathway was reinforcing, because the vSub projects to the NAc and because the stimulation of this pathway has been shown reinforcing. Our data do not support such a hypothesis; however, they were sufficient to conclude that the stimulation of SuM:vSub is not reinforcing in the way we tested.

5. The EPSPs in TD-tomato positive neurons in Figure 4g looked much more equivocal than the text suggested. The “representative” plot in 4f does not look very representative of the bar plot, where there only appears to be 1 highly activated cell and several around zero.

The authors' response:

This experiment was to test the hypothesis that SuM GLU neurons form functional synapses with MS neurons, with particular interest in the MS GLU neurons. As for the highly activated cell, it may be that there is a small population of neurons strongly driven by that input, but that the majority of the connections are not as robust. If the current proportions can be translated to the wider cell population in the septal area then about 15-20% of the connected cells may potentially be strongly connected (given that 1 of 6 cells are strongly connected). In general, there is much variability in these experiments due to variable levels of ChR2 terminal expression and tDTomato transfection, etc. While an oEPSP amplitude of 2-5mV (the range of most of the cells) is relatively small, it still indicates functionality of the pathway. Despite the magnitude of oEPSP, all oEPSP amplitudes were greatly reduced by GLU antagonism. We also now provide additional analysis showing short latency of EPSP to laser onset as additional evidence of SuM GLU neurons ability to drive MS neurons, including MS GLU neurons. As for the representative traces, they are averaged responses from the cell that is in the midpoint of the data sets. We have clarified how the represented trace was created in the figure legend.

6. It wasn't clear to me at what point during the operant sucrose seeking task the recordings were taken. This is important given the animals' behavior was changing across the sessions.

Depending on the session, the data time-locked to entering the sucrose port after the CS- might mean very different things.

The authors' response:

Prior to the recording session, animals were extensively trained until they displayed stable responding between sessions. We have added more details to the methods to make it clearer (see page 36-37).

Minor comments

- The labeling of figure 2 does not match the main text.

The authors' response:

We have replaced the labeling with that matching the text.

- Please somewhere note numbers of neurons recorded / animal.

The authors' response:

Thank you, we have added this information to the methods section (page 34).

- Why were the dopamine antagonist doses not counterbalanced across animals?

The authors' response:

The aim of the experiment was to examine whether the two groups responded differently to the dopamine receptor blockade, but not how each dose affects behavior. To this end, we tested the two groups with an ascending order of doses to minimize possible variability caused by potential varying carry-over effects by testing a high-dose antagonist first with a counterbalancing test.

- P32 – say how much the laser power was increased / decreased for the mice in the SuM:Sept and VTA:VStr mice

The authors' response:

Laser intensities were equal for both groups (~10mW) during acquisition. Intensities were increased to up to 30mW for mice with response rates less than 200 press/session (mainly VTA:VStr group) and decreased down to 3-5mW for mice with response rates greater than 200 press/session (mainly SuM:Sept group).

- There are many more refined protocols for rat anesthesia than the one used here, particularly given the potential for irritation of chloral hydrate administered ip. I would advise the group to update their anesthetic regime for rats if they haven't already done this. Also, can the authors please confirm that their local IACUC approved this regime.

The authors' response:

We are aware of the movement of the animal care and use community to discourage the use of chloral hydrate-based anesthetics such as Equithesin. We no longer use it. The rat experiment described in the manuscript was conducted a few years ago, and the experiment was approved by

the NIDA-IRP ACUC prior to the experiment. To date, our institution's NIDA-IRP ACUC has not banned the use of Equithesin, although it strongly discourages the use and requires scientific justification for the use. Despite some occasional side-effects, Equithesin had been the first choice for many labs in the past. We assure you that these rats were healthy at the time of testing.

- P5, line 155, F or p value must be wrong

The authors' response:

The F value is wrong. The typo is now corrected. Thank you for identifying the mistake.

- Is the fMRI experiment really just a "pilot" experiment (p11, line 327)?

The authors' response:

We removed the word pilot. We failed to delete it after extensive analysis of the fMRI data.

Reviewer #3 (Remarks to the Author):

In this study Kesner et al. are reporting a careful analysis of the role of the supramammillary nucleus (SuM) and glutamatergic projections of this nucleus on operant reward conditioning of mice. Although highly interesting and relevant for the field of circuit neuroscience, the organization and presentation of the results appear not as rigorous as the realization of the experiments. Moreover the ex vivo physiology experiment do not support the conclusions of a monosynaptic connection as claimed by the authors.

The authors' response:

We have re-written the entire manuscript, to make it easier to read through. In addition, we provide additional analysis for the ex vivo physiology experiment.

First, the title, abstract and results state a role for SuM neurons in anticipatory behaviors. However, the experiments are very specific of reward anticipation, and the selectivity of this study for positive valence should be clarified starting in the title of the manuscript.

The authors' response:

We have revised the title. We replaced "anticipatory behavior" with "approach behavior", which is reward-anticipation behavior. We agree that the term anticipatory is confusing because it implies cognition while the role of the SuM is motivation or arousal rather than cognition. In addition, we have extensively re-written the entire manuscript, to clarify the rationale, aims and interpretations of the data.

Second, the Figure 6 describing general inhibition of the SuM (no projection specificity) would allow a better narrative if it was the second Figure, before analysis of the influence of different projections of the SuM.

The authors' response:

We appreciate the reviewer's advice on reorganizing the manuscript to improve the narrative. While we left the order of experiments unchanged, we significantly rewrote sections of the manuscript to improve the flow of the narrative.

Third, in the ex vivo electrophysiological experiments, the authors claim they identified a monosynaptic connection. However, they did not block recurrent excitation and the current they observe in baseline condition (ACSF) might be due to polysynaptic excitation. To unequivocally demonstrate a monosynaptic connection, the authors should use ChR2-assisted circuit mapping (CRACM) by applying TTX and 4AP on the sample.

The authors' response:

We have provided an additional analysis of the latency to postsynaptic response onset. The average latency to response onset is ~0.92 milliseconds, which is in line with previous results showing a response latency in cortical rat neurons of approximately 1.1 milliseconds (Frick et al 2008).

Finally, from the in vivo electrophysiology recordings (tetrodes, Figure 5), the author conclude that the SuM neurons are inactive during reward consumption. However, the data represented are changes in firing rate, and a decrease of firing does not necessarily reflect that the neurons are inactive. The authors should either rephrase their conclusions, or represent the absolute firing rates in the figure, rather than z-scores.

The authors' response:

We have re-phrased the description of neural activity by replacing the term “inactive” with “decrease”.

--- Minor Comments -----

The authors seem to interchange reward and reward seeking, which is inaccurate and confusing. In the abstract the author mention “SuM-Septum reward”, when what they mean is likely reward seeking, or another reward-related behavior. The reward is the reinforcer (sucrose consumption, or optogenetic manipulation of a neuron population) not the behavioral response or state. Similarly in the introduction (line 44) the authors say “Such brain manipulations must elicit a transient state, referred to as reward, that triggers goal-directed information processing for

foraging or other seeking behavior; that is, anticipatory behavior.” This definition attempt is very unclear. Do the authors mean ‘reward state’ rather than reward (which is the reinforcer) ? or maybe they mean reward expectation ? reward anticipation ? It is crucial to correct and clarify these definitions.

The authors’ response:

We have realized that defining reward as such state is confusing. We have re-written the entire introduction, to describe our perspective in a clearer manner and replaced confusing terminology with clearer one.

The brain schematic representing the location of viral injection and optic fiber placements are hardly readable as they are very small (Figure 1A, 2A, 2E...). An expanded schematic (or just with 3 boxes: SuM, septum, vSub) would facilitate understanding of the experimental design.

The authors’ response:

We have done our best to expand these graphics were possible and make sure the experiment preparations are clear throughout the text, figures and legends, and methods.

Numbering the experiments in the titles of the sections is not informative and distracting for the reader. Please remove this part of the title.

The authors’ response:

We removed these from the section titles and reworded references to specific experiments in the text.

Figure 1 H: the author mention 50 nL in the Figure, but the meaning (volume of viral vector injected) is not explained in the legend. Please specify this in the legend.

The authors’ response:

We now describe this group in the legend.

The font of the figures is sometimes Arial and sometime Calibri. Aesthetic would be better with a uniform font.

The authors’ response:

We thank the reviewer for their careful observations on consistency and overall aesthetic. We revised to keep uniform font.

REVIEWER COMMENTS

Reviewer #1 (Remarks to the Author):

This impressive series of experiments make a significant step toward resolving the functional role of the SuM and its circuits. Specifically, they show that activation of glutamate (not GABA/glut nor DA) SuM neurons is reinforcing, and their activity is associated with appetitive but not consummatory behavior. This SuM-MS pathway seems to depend on dopamine and also drives the VTA-vStr DA pathway. Interestingly, the activation of SuM terminal in the PV thalamic neurons results in aversive behavior. These findings are discussed in terms of the extant literature that describes the role of the SuM as being associated with learning, theta activity, reward, and arousal. This paper is important in helping make sense of how these functions of the SuM are related.

Overall, this is an important study that is of wide interest, given the explanation of SuM functions in terms of a literature that ascribes apparently disparate roles. The paper is well-written, very clear, the argument is logical and there are few problems, and a few opportunities for further work that would not sensibly belong in this already large work.

Injection sites are large and use a needle, but fiber locations are plotted in supplementary material to validate to site of stimulation or photometry.

Major:

1. The channelrhodopsin-2-assisted circuit mapping latency is very short at about 1 ms. The citation for this duration is inappropriate, from paired recordings. Typically, latencies of CRACM are slightly longer than electrical stimulation. I would recommend a different citation (e.g. Petreanu ... Svoboda, 2007, or other) and a description of how this might be the case.
2. Holding potentials for patch-clamp were not described, and IVs not performed to see if there was also a chloride conductance from glut/GABA neurons of the SuM. Please at least mention the chloride reversal and holding potentials and whether an IV was performed after DNQX/AP5.

Minor:

2. Page 4, line 74, write intracranial self-stimulation in full before ICSS.
3. Page 4, para 1-2: Mention that VTA stimulation was unilateral.
4. Why is the anterior thalamus activated with SuM-MS activation and not VTA-vStr if the former pathway activates the latter? Please at least briefly discuss this.
5. Page 12, line 417. Perhaps citation missing? "... arousal ^interaction^".

Nigel Pedersen

Reviewer #2 (Remarks to the Author):

The authors have generally done a good job at revising the manuscript to tighten up the terminology and to improve the flow of the paper. The new fiber photometry data is also a useful addition (though a few more details would be helpful – for instance, it wasn't entirely clear to me if the SuM  septum stimulation + photometry recordings were done in awake animals?). I was still somewhat unconvinced about how the electrophysiology data fit with the story or that there were tight enough controls on the behavior to reach strong conclusions about what processes were being affected. However, overall, I was persuaded that the data are sufficiently novel and robust to be of real interest to the field.

Reviewer #3 (Remarks to the Author):

The revised manuscript has slightly improved in terms of logical flow, concepts and rational. The authors addressed most of my comments but one remains unanswered and it is a very important issue.

In my first review I mentioned that the only complete demonstration of a monosynaptic connection is to perform ex vivo circuit mapping with optogenetics (CRACM). The authors did not perform these experiments in the year they took to submit their revised manuscript. However, they now provide latencies from light offset to optogenetic-EPSC onset. While latencies can be informative and could be convincing, I have *ethical concern* regarding their quantification.

Indeed, the average latencies in Figure 4L are below 1 ms, but when looking at the example traces, this does not appear possible: when measuring the latencies on the traces of Figure 4i, they are of 4.32 ms for the TdT- and 7.23 ms for the TdT+ cells (please see enclosed file), while all individual data of Figure 4L are below 1.3 ms.

I initially thought that their 50 ms time scale is wrong as with this scale, the opto stim would be about 5 ms, and it is mentioned to be 2 ms in the methods. However, even assuming the light stim is 2 ms (the scale bar would then be ~18 ms), latencies are 1.54 ms for the TdT- and 2.58 ms for the TdT+ cells, which is still above all single data points represented in Figure 4L.

Therefore, I do not believe the data represented in Figure 4, which is also questioning the rest of the findings in this manuscript in my opinion.

Specific comments -----

Abstract

Line 22: replace illuminate with elucidate

Line 23: add 'Using optogenetics, we found that...'

Introduction

Line 63: 'that suggest reinforcing effects are induced by the stimulation of SuM glutamatergic (GLU) neurons projecting to MS GLU neurons and MS GLU neurons projecting to the VTA'.

This sentence is not accurate. Indeed, the authors did not demonstrate that the SuM-MS-VTA pathway is responsible for *all* reinforcing effects.

Please rephrase with 'suggesting that the SuM-MS-VTA pathway can induce reinforcement'.

Results

Line 276: remove the comma

Line 311: replace Th with TH

Figure 2A: the graph is very confusing: to help please make the dots of one color for stimulation of one region (even if they are not on the same curve/same animals)

The authors improved the size of their experimental diagrams, but they are still very hard to decipher. I strongly advise the authors to replace them with simpler schematic diagrams (one box of a specific color representing each brain regions). Please see attached file (page 2).

Figure 4F: opto-EPSC latencies

Figure 1a

Figure 2a

Shown below are our point-to-point responses (in sky blue) to the reviewers' comments (in black). The changes made in the manuscript are shown in emerald green texts.

REVIEWER COMMENTS

Reviewer #1 (Remarks to the Author):

This impressive series of experiments make a significant step toward resolving the functional role of the SuM and its circuits. Specifically, they show that activation of glutamate (not GABA/glut nor DA) SuM neurons is reinforcing, and their activity is associated with appetitive but not consummatory behavior. This SuM-MS pathway seems to depend on dopamine and also drives the VTA-vStr DA pathway. Interestingly, the activation of SuM terminal in the PV thalamic neurons results in aversive behavior. These findings are discussed in terms of the extant literature that describes the role of the SuM as being associated with learning, theta activity, reward, and arousal. This paper is important in helping make sense of how these functions of the SuM are related.

Overall, this is an important study that is of wide interest, given the explanation of SuM functions in terms of a literature that ascribes apparently disparate roles. The paper is well-written, very clear, the argument is logical and there are few problems, and a few opportunities for further work that would not sensibly belong in this already large work.

Injection sites are large and use a needle, but fiber locations are plotted in supplementary material to validate to site of stimulation or photometry.

Major:

1. The channelrhodopsin-2-assisted circuit mapping latency is very short at about 1 ms. The citation for this duration is inappropriate, from paired recordings. Typically, latencies of CRACM are slightly longer than electrical stimulation. I would recommend a different citation (e.g. Petreanu ... Svoboda, 2007, or other) and a description of how this might be the case.

The authors' response: We have added detail of how we measured the latency in our slice-electrophysiology experiments to the methods section. We also added a few sentences in the results making note of our fast latency observation and why it appears to be faster than the Petreanu et. al. study (page 7, lines 220-228). Thank you for suggesting the alternate citation for this piece of our data.

2. Holding potentials for patch-clamp were not described, and IVs not performed to see if there was also a chloride conductance from glut/GABA neurons of the SuM. Please at least mention the chloride reversal and holding potentials and whether an IV was performed after DNQX/AP5.

The authors' response: Our patch clamp slice physiology experiments were performed in current-clamp mode so there is no holding potential to report.

Minor:

2. Page 4, line 74, write intracranial self-stimulation in full before ICSS.

The authors' response: Thank you, added.

3. Page 4, para 1-2: Mention that VTA stimulation was unilateral.

The authors' response: Now noted at first instance of introducing VTA site control group (page 4, line 77).

4. Why is the anterior thalamus activated with SuM-MS activation and not VTA-vStr if the former pathway activates the latter? Please at least briefly discuss this.

The authors' response: Sorry for a confusion. The anterior thalamus was activated by the VTA-VStr stimulation, but not by the SuM-MS stimulation. The SuM-MS stimulation did activate the VStr although the activation was moderate. The moderate effect of the SuM pathway on VStr signals may be explained by the fact that fMRI was performed in anesthetized mice.

5. Page 12, line 417. Perhaps citation missing? "... arousal ^interaction^".

The authors' response: We cited work that contributed to this reasoning.

Nigel Pedersen

The authors' response: We thank the reviewer for the careful and thoughtful comments, which made the study more interesting and thorough.

Reviewer #2 (Remarks to the Author):

The authors have generally done a good job at revising the manuscript to tighten up the terminology and to improve the flow of the paper. The new fiber photometry data is also a useful addition (though a few more details would be helpful – for instance, it wasn't entirely clear to me if the SuM  septum stimulation + photometry recordings were done in awake animals?). I was still somewhat unconvinced about how the electrophysiology data fit with the story or that there were tight enough controls on the behavior to reach strong conclusions about what processes were being affected. However, overall, I was persuaded that the data are sufficiently novel and robust to be of real interest to the field.

The authors' response: We have clarified that the fiber photometry was performed in awake freely moving animals (page 10, line 372). We thank the reviewer for the thoughtful comments during this process.

Reviewer #3 (Remarks to the Author):

The revised manuscript has slightly improved in terms of logical flow, concepts and rational. The authors

addressed most of my comments but one remains unanswered and it is a very important issue.

In my first review I mentioned that the only complete demonstration of a monosynaptic connection is to perform ex vivo circuit mapping with optogenetics (CRACM). The authors did not perform these experiments in the year they took to submit their revised manuscript. However, they now provide latencies from light offset to optogenetic-EPSC onset. While latencies can be informative and could be convincing, I have *ethical concern* regarding their quantification.

Indeed, the average latencies in Figure 4L are below 1 ms, but when looking at the example traces, this does not appear possible: when measuring the latencies on the traces of Figure 4i, they are of 4.32 ms for the TdT- and 7.23 ms for the TdT+ cells (please see enclosed file), while all individual data of Figure 4L are below 1.3 ms.

I initially thought that their 50 ms time scale is wrong as with this scale, the opto stim would be about 5 ms, and it is mentioned to be 2 ms in the methods. However, even assuming the light stim is 2 ms (the scale bar would then be ~18 ms), latencies are 1.54 ms for the TdT- and 2.58 ms for the TdT+ cells, which is still above all single data points represented in Figure 4L.

Therefore, I do not believe the data represented in Figure 4, which is also questioning the rest of the findings in this manuscript in my opinion.

The authors' response: We thank the reviewer for noticing discrepancies between Figure 4i and 4k and took time and effort to make points. We completely agree that the panel shown in Figure 4i is confusing. We made two alterations. First, the pulse illustration is now scaled as close to as relatively correct length. We meant the panel to be schematic rather than actual. Second, we now clearly describe that traces are average responses. We described the traces as “representative EPSPs” in the previous manuscript. We have revised the panel and rewrote the legend.

In addition, we now provide a more complete description for how the latency measurement was calculated in the methods section (page 34, lines 942-946). For your review, here is a detailed account from the slice electrophysiologist who conducted this experiment on how they performed the latency analysis:

“For latency analysis, I first convert WCP files that are output from WinWCP acquisition software into ABF files. I then use Axograph software to measure response latency. I average 5-10 sweeps from the baseline period together to provide a cleaner (less noisy) response to use for latency measurement. Then, I use threshold detection in Axograph to detect event onset latency as this provides the most accurate measurement of onset latency and corresponds to what visually appears to be the first light-mediated deflection in membrane potential. The slope of this can vary from episode to episode, but is in the range of 1-5 mv/s. If it appears that the event detection has failed to capture the correct window, I will then use manual detection to identify the response onset. Axograph outputs a value in a log file for the event onset based on these detection parameters. Using this methodology, response onset was very fast, typically at or below a millisecond for the first voltage deflection after light onset.”

Specific comments -----

Abstract

Line 22: replace illuminate with elucidate

The authors' response: We have made this change.

Line 23: add 'Using optogenetics, we found that...'

The authors' response: Thank you, we have added this to the abstract.

Introduction

Line 63: 'that suggest reinforcing effects are induced by the stimulation of SuM glutamatergic (GLU) neurons projecting to MS GLU neurons and MS GLU neurons projecting to the VTA'.

This sentence is not accurate. Indeed, the authors did not demonstrate that the SuM-MS-VTA pathway is responsible for *all* reinforcing effects.

Please rephrase with 'suggesting that the SuM-MS-VTA pathway can induce reinforcement'.

The authors' response: Thank you. We have revised the sentence accordingly.

Results

Line 276: remove the comma

Line 311: replace Th with TH

The authors' response: Thank you for both edits. We've made the changes.

Figure 2A: the graph is very confusing: to help please make the dots of one color for stimulation of one region (even if they are not on the same curve/same animals)

The authors' response: We have edited the graph accordingly. We also adopted your suggested schematics for Figure 2a, to clarify the injection and stimulation sites. Thanks!

The authors improved the size of their experimental diagrams, but they are still very hard to decipher. I strongly advise the authors to replace them with simpler schematic diagrams (one box of a specific color representing each brain regions). Please see attached file (page 2).

The authors' response: Thanks for taking time and preparing the schematics in the attached file. We adopted them for Figures 1a and 2a, to clarify the injection and stimulation sites. The careful and thoughtful reviews during this process have greatly improved the manuscript.

REVIEWERS' COMMENTS

Reviewer #3 (Remarks to the Author):

The authors addresses my main concerns and the study is providing very interesting results for our understanding of hypothalamus function and organization.

Some improvements on the representation of the experiments and results can still be made:

-for the representation of the surgical procedure, please remove all the schematics of the brains (sagittal section) which are still hardly readable, and keep only the boxes representations which are much clearer (boxes are still missing in some figures (1f, 4, 7 and 8)).

-in Figure 1a the authors wrote MM instead of MB. Please add OR between each brain regions, as suggested in my previous review. Attached is an alternative representation which would be even clearer. Please consider including it.

-for all Figures, the mouse line should appear first (before any diagrams), i.e. on the top left of the panels, not bottom right.

Figure 1a

Wild type mice

AND

SuM:SuM

OR

SuM:MB

OR

SuM:VTA

Response to the reviewers' comments

Shown below are our point-to-point responses (in maroon) to the reviewers' comments (in blue).

Reviewers' comments:

Reviewer #3 (Remarks to the Author):

The authors addresses my main concerns and the study is providing very interesting results for our understanding of hypothalamus function and organization.

Some improvements on the representation of the experiments and results can still be made:

-for the representation of the surgical procedure, please remove all the schematics of the brains (sagittal section) which are still hardly readable, and keep only the boxes representations which are much clearer (boxes are still missing in some figures (1f, 4, 7 and 8)).

The authors' response:

We have replaced the sagittal sections with much simpler ones and provided box representations for the figures. Thanks!

-in Figure 1a the authors wrote MM instead of MB. Please add OR between each brain regions, as suggested in my previous review. Attached is an alternative representation which would be even clearer. Please consider including it.

The authors' response:

We have corrected the error and added "OR" to the figures according to your suggestion. Thanks.

-for all Figures, the mouse line should appear first (before any diagrams), i.e. on the top left of the panels, not bottom right.

The authors' response:

We have made the modification. Thanks!